# Englacial Architecture of Lambert Glacier, East Antarctica

Rebecca J. Sanderson[1], Kate Winter[2], S. Louise Callard[1], Felipe Napoleoni[3,4], Neil Ross[1], Tom A. Jordan[5], Robert G. Bingham[6]

[1]School of Geography, Politics and Sociology, Newcastle University, Newcastle, UK

[2]Department of Geography and Environmental Sciences, Faculty of Engineering and Environment Northumbria University, Newcastle, UK

[3]Department of Geography, Durham University, Durham, UK

[4]Centro de Estudios Científicos, Valdivia, Chile

[5]British Antarctic Survey, Cambridge, UK

[6]School of GeoSciences, University of Edinburgh, Edinburgh, UK

*Correspondence to*: Corresponding author: Rebecca Sanderson (r.sanderson5@newcastle.ac.uk)

**Abstract.** The analysis of englacial layers using radio-echo sounding data enables the characterisation and reconstruction of current and past ice-sheet flow. Despite the Lambert Glacier catchment being one of the largest in Antarctica, discharging ~16% of East Antarctica's ice, its englacial architecture has been little analysed. Here, we present a comprehensive analysis of Lambert Glacier's englacial architecture using radio-echo sounding data collected by the Antarctica's Gamburtsev Province Project (AGAP) North survey. We used an internal-layering continuity index (ILCI) to characterise the internal architecture of the ice and identify four macro-scale ILCI zones with distinct glaciological contexts. Whilst the catchment is dominated by continuous englacial layering, disrupted or discontinuous layering is highlighted by the ILCI at both the onset of enhanced ice flow (defined here as >15 ma$^{-1}$) and along the shear margin, suggesting a transition in englacial deformation conditions and converging ice flow. These zones are characterised by buckled and folded englacial layers which have fold axes aligned with the current ice-flow regime. These folds suggest that the flow direction of the Lambert Glacier trunk has changed little, if at all, during the Holocene. Disturbed englacial layers that do not correspond to modern ice-flow routing found within a deep subglacial channel, however, suggests that ice-flow change has occurred in a former tributary that fed Lambert Glacier from grid north. As large outlet systems such as Lambert Glacier are likely to play a vital role in the future drainage of the East Antarctic Ice Sheet, constraining their englacial architecture to reconstruct their past ice flow and determine basal conditions is important for refining projections of future sea level change.

## 1 Introduction

The Antarctic Ice Sheet, particularly in West Antarctica, is currently undergoing rapid thinning (Fricker et al., 2012; Shepherd et al., 2018; Rignot et al., 2019). Whilst changes in East Antarctica have been less widespread and lower in magnitude than those observed in West Antarctica (Gardner et al., 2018), East Antarctica accounts for ~75% of the total Antarctic Ice Sheet area and contains 52 m of potential sea-level rise (Fricker et al., 2021; Stokes et al., 2022). As temperatures increase, major outlet glaciers in East Antarctica could see increases in ice discharge and changes to ice-flow (Li et al., 2016; Miles et al., 2021; IPCC, 2021), with subsequent impacts for future sea-level rise (Miles et al., 2013; Stokes et al., 2022). To project future sea-level rise, empirical constraints are required to inform ice-sheet models. These constraints include regional flow history, ice-sheet structure and rheology, and basal boundary conditions, and can be derived from radio-echo sounding (RES) characterisation of the englacial architecture and bed of the ice sheet (Holschuh et al., 2017).

Lambert Glacier is part of the wider Lambert-Amery system (Fig. 1a) (Fricker et al., 2000), the largest glacier-ice-shelf system in East Antarctica, making it an important component of overall ice-sheet mass-balance. Mass-balance observations (Wen et al., 2008; Yu et al., 2010) and numerical-modelling studies (Bassis et al., 2005; Glasser et al., 2015) suggest that the Lambert-Amery system currently has an overall positive mass balance of 20.9 ± 1.9 Gt a$^{-1}$ (Gong et al., 2014; Pittard et al., 2017; Cui

et al., 2020), and significant future retreat beyond the present-day grounding line is not projected over the next 500 years, even under a range of future climate scenarios (Pittard et al., 2017). However, low-angle retrograde slopes 250 km upstream of Lambert Glacier's current grounding line mean that marine-ice-sheet instability may occur over longer timescales (Gong et al., 2014). A deep trough situated below Lambert Glacier extends ~600 km inland of the grounding line. Subglacial troughs such as these have the ability to drain the ice sheet rapidly (Morlighem et al., 2020). Because of this potential, and also because areas of Antarctica which have been "stable" for a long time can hold vital information about ice-sheet history (Pattyn et al., 2020), this study characterises the flow of Lambert Glacier from analysis of the catchment's englacial stratigraphy and architecture in RES profiles.

Englacial features (e.g. englacial layers and basal ice units) identifiable in RES data (e.g. Siegert, 1999; Bingham et al., 2007a; Bell et al., 2011; Schroeder et al., 2020) record past and present ice flow and internal deformation. The shape, form, presence, and even absence, of radar-imaged englacial features have been used to reconstruct past disruptions in flow regime (e.g.Siegert et al., 2003b; Bingham et al., 2007b) and accumulation rates (e.g. Cavitte et al., 2018; Winter et al., 2019; Ashmore et al., 2020; Bodart et al., 2021); whilst also allowing us to incorporate age tracers into ice-sheet models (e.g. Parrenin et al., 2017; Sutter et al., 2021). In this study, we undertake analysis of the englacial architecture of Lambert Glacier using the Internal Layering Continuity Index (ILCI; after Karlsson et al., (2012); Bingham et al.,(2015)) to assess the current flow regime, and better understand past changes in the region, which could elucidate future change.

## 2 Study Area

Lambert Glacier (71–75° S, 68° E) contains ~1,480,000 km$^2$ of ice, comprising 16% of the grounded ice of East Antarctica (Fricker et al., 2000), equivalent to ~8 m of global sea level (Rignot et al., 2019). With a width of over 40 km and a length of 400 km from the grounding line to the southernmost point in which ice flow is visible on Landsat optical imagery, Lambert Glacier is the largest ice stream in the world (Allison, 1979). Draining into Amery Ice Shelf, ice flow in the Lambert Glacier catchment increases away from the ice divide at Dome A, where ice flow is 0-5 ma$^{-1}$ (Mouginot et al., 2019) (Fig. 1c). The onset of enhanced ice flow is defined as the location of the transition between inland ice flow and streaming ice flow (Bindschadler et al., 2001); in Lambert Glacier we define this transition where ice velocity exceeds 15 ma$^{-1}$. The BedMachine (v2) compilation of basal topography in Antarctica (Morlighem et al., 2020) (Fig. 1b) shows that this enhanced ice flow is channelised through a deep topographic graben-like rift valley depression, known as the Lambert Rift (Leitchenkov et al., 2018). The Lambert Rift has a topographic relief of ~3500 m (Fig. 1b), with the bed resting >2000 m below sea-level (Fig. 1b) (Allison, 1979; Morlighem et al., 2020). Where the rift valley is deepest, the ice is >4 km thick (Morlighem et al., 2020) (Fig. 1d). Ice-flow velocity increases markedly (i.e. from 15 to 250 ma$^{-1}$) as ice flows through the rift, towards the Amery Ice Shelf (Fig. 1c, 1d).

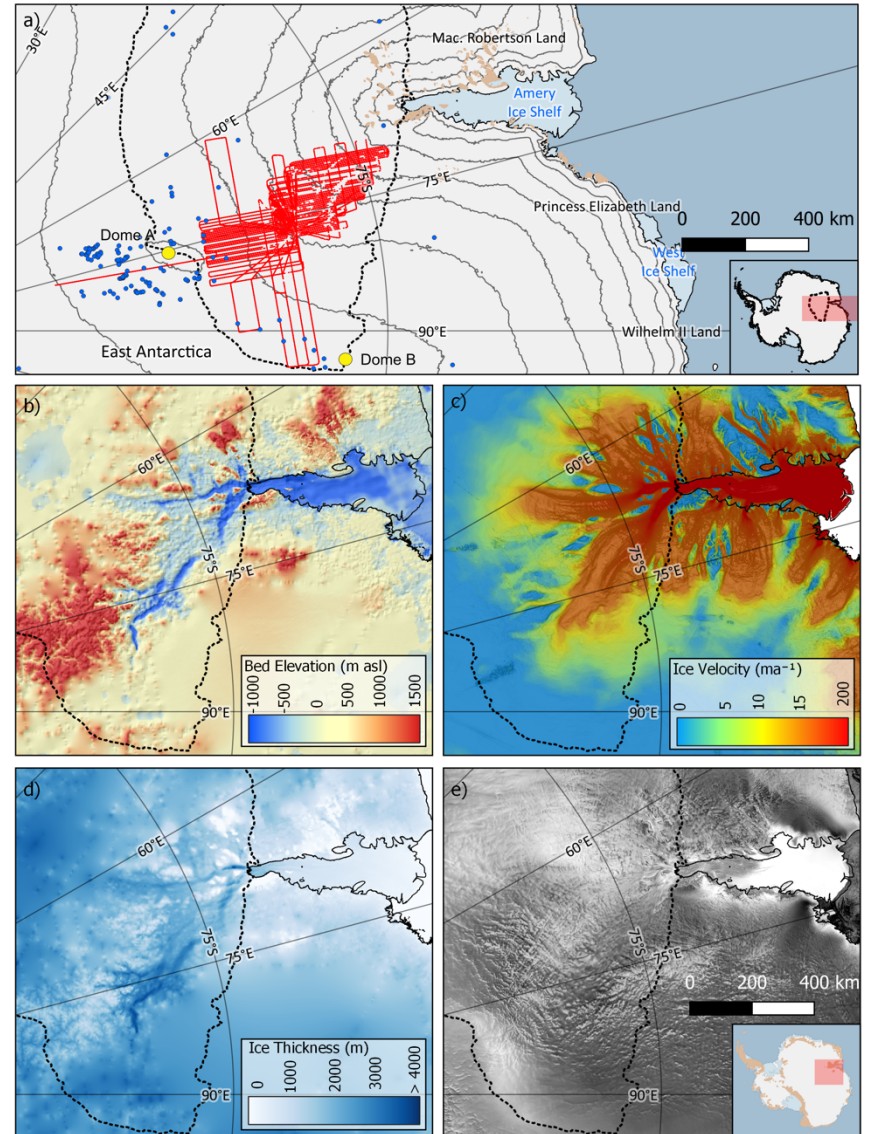

**Fig. 1. Location of Lambert Glacier, within the wider Lambert-Amery system of East Antarctica. The Lambert-Amery catchment is marked with a dotted black line (Zwally *et al.*, 2012b). (a) Reference Elevation Model of Antarctica (REMA)-derived ice-surface elevation showing the location of relevant AGAP-N RES flight lines (red). Contours (grey solid line) are at 500 m intervals (Howat *et al.*, 2019). Dome A and B are highlighted by yellow circles. Subglacial-lake locations (Livingstone *et al.*, 2022) are marked with blue circles. (b) Bed elevation of the study area from BedMachine v2 (Morlighem *et al.*, 2020), which includes the AGAP-N RES bed-elevation data. The grounding line is shown with a black line (Bindschadler, *et al.*, 2011). (c) Ice-flow speed averaged over 1992-2017, overlying a raster derived from calculating the slope from the maximum horizontal gradient of ice velocity. This underlying raster has a grey-scale colour pallet and is histogram equalised with the colour scale saturated at 65 ma$^{-1}$ (Mouginot et al., 2019). (d) Ice thickness from BedMachine v2 (Morlighem *et al.*, 2020). (e) High resolution RAMP RADARSAT-1 radar imagery of Lambert Glacier (Jezek, 1999)**

## 3 Methods

### 3.1 Data

The RES data used for this study were acquired in 2007-2009 over the northern region covered by the internationally collaborative aerogeophysical survey of Antarctica's Gamburtsev Province (AGAP) Project (Bell et al., 2011; Ferraccioli et al., 2011; Rose et al., 2013). The total "AGAP North" survey grid (highlighted in Fig. 1a) comprises 120,000 line km of data, with an 'across line' spacing of 5 km and 'tie lines' ~33 km apart.

All data were acquired with the 150 MHz British Antarctic Survey PASIN (Polarmetric Airborne Survey INstrument) radar
system, operating over a bandwidth of 15-20 MHz, and deploying a 4 μs, 10MHz linear chirp designed to sound deeply into the ice. The radar system has a vertical resolution of ~8.4 m and a horizontal trace spacing (post-processing) for the AGAP survey of ~20 m. 2-D Synthetic Aperture Radar (SAR) processing was applied to enhance resolution and increase signal-to-noise (Frémand et al., 2022). Previous applications of these data have: (i) used bed echoes to map subglacial topography (Fretwell et al., 2013); (ii) explored the geological and geomorphological evolution of the Gamburtsev Subglacial Mountains
(Ferraccioli et al., 2011; Rose et al., 2013); and (iii) analysed the region's distinctive basal ice units (Bell et al., 2011). Across part of the same region, Luo et al. (2020) used a single traverse of ground-based ice-penetrating radar data from Zhongshan station to Dome A to investigate ice-sheet internal structure and basal roughness, applying similar analysis to that deployed for this paper, but over a much more geographically-limited area. Our study is the first to interrogate comprehensively the englacial architecture of Lambert Glacier catchment using multiple radar survey lines. To place our analysis and findings in a
wider context, we combine our analysis of the radar data with freely available MEaSUREs ice-surface velocity data (Mouginot et al., 2019), BedMachine bed-topography maps (Morlighem et al., 2020) and RADARSAT Antarctic Mapping Project RAMP satellite imagery (Jezek, 1999).

### 3.2 Internal layer continuity analysis

Following Karlsson et al. (2012) and Bingham et al. (2015), we applied the "Internal Layer Continuity Index" (ILCI) to classify
and map englacial stratigraphy across the Lambert Glacier catchment. The ILCI was initially developed as a tool for objectively and quantitatively capturing the degree to which continuous and flat isochrones had previously been qualitatively observed to be buckled or disrupted by variations in ice flow (e.g. Rippin et al., 2003; Siegert et al., 2003b; Bingham et al., 2007b; Karlsson et al., 2009). The ILCI operates by quantifying the average power of peaks observed through the ice column in each trace, and then further averaging the trace-by-trace results over windows of a predefined length along radar tracks (Fig. 2). This method
employs minimal processing to avoid any dependency on filtering and background modification which may modify the englacial layer visibility. Although the ILCI does not directly analyse the continuity of high-power peaks between adjacent traces, the most commonly accepted explanation for consistently high ILCI measured across multiple kms of ice is that englacial layering is continuous and undisturbed over that region, as noted in several previous studies (e.g. Karlsson et al., 2012; Karlsson et al., 2014; Bingham et al., 2015; Winter et al., 2015; Winter et al., 2016). The most common explanation for

low ILCI across a region is that the reflection power (as measured trace by trace) has been reduced by buckling, folding, warping, or even the complete absence of englacial layering. This disruption is understood to be the result of ice currently, or previously, experiencing significant deformation as it flows through an ice-stream onset or shear zone, or around/over significant subglacial protuberances (Bingham et al., 2015; Winter et al., 2015; Winter et al., 2016). Previous applications of ILCI have noted that buckling/folding might be missed if the radar lines parallel the fold axes (see depiction in Ng and Conway,

2004), which is typically where the radar lines have been gathered along or close to the ice-flow direction (Karlsson et al., 2012; Bingham et al., 2015). However, in practice very few airborne radar lines in Antarctica (including those analysed in this study) have been acquired sufficiently close to the alignment of ice-flow for this effect to dominate ILCI interpretations. ILCI therefore provides a useful mechanism for reconnoitring englacial conditions across large regions of an ice sheet, to characterise broad ice-flow conditions and guide more detailed investigations in regions of interest (c.f., Frémand et al., 2022).

It is in this spirit that we apply the ILCI technique in this study.

Following previous ILCI applications (Karlsson et al., 2012; Bingham et al., 2015; Winter et al., 2016), the upper and lower 20% of the ice column was removed to avoid ILCI bias. This occurs where englacial layers are often absent in RES returns from deep in the ice sheet (lowermost 20%) and where surface clutter (upper 20%) may impact the results. We averaged ILCI

over windows of 100 and 1000 traces (2 -20 km along track) to identify both small-scale 'localised' anomalies and more regional trends. From the broad range of ILCI values retrieved by ILCI across Lambert Glacier, we define a "high-continuity" ILCI value to be >0.45. This value is higher than other studies which have applied ILCI to PASIN radar data which have used 0.18 (Karlsson et al., 2012), 0.10 (Bingham et al., 2015), and 0.06 (Winter et al., 2015; Winter et al., 2016) as an indicative threshold after which continuity is defined as "high". However, our use of a 0.45 threshold is consistent with the

recommendation from a recent pan-Antarctic application of ILCI that the threshold will vary depending on the purpose of the study and the region of interest (Frémand et al., 2022). The "high" ILCI value defined over the Lambert Glacier catchment reflects large variation in analysis output when applied to a larger region, like the one in this study, which contains significant variations in bed elevation and ice velocity.

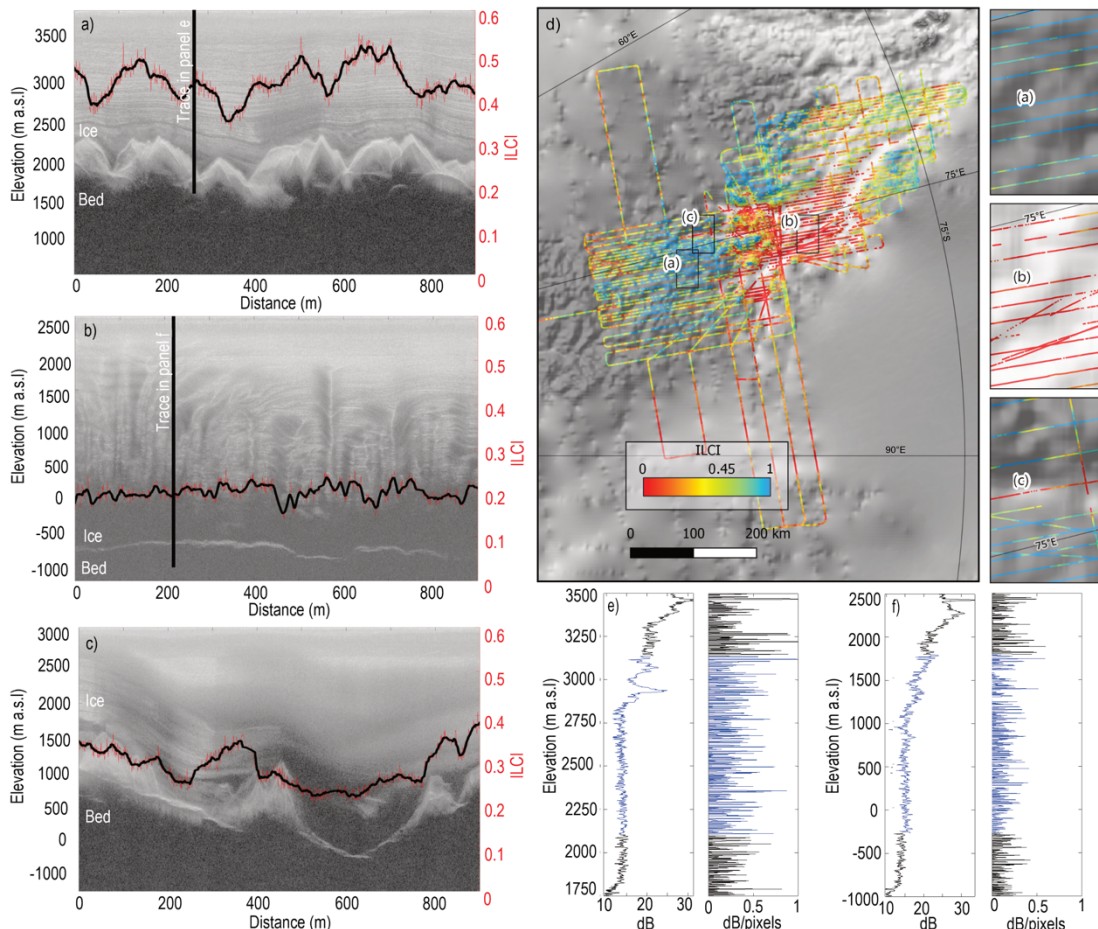

Fig. 2. Example radargrams from the AGAP North survey showing: (a) continuous englacial layers along RES survey line A44b, (b) discontinuous layering within RES survey line A29h and (c) absent layers in RES survey line A39d. In each panel, the continuity index (ILCI) is shown in red (unsmoothed), with a running mean of 10 traces highlighted by the black line. (d) Profile locations marked onto site-wide ILCI results (using a running mean of 100 traces). Background is BedMachine v2 bed topography (Morlighem *et al.*, 2020). (e) A-scope plot demonstrating relative received power (dB) (left) and the absolute value of the gradient, for a trace in panel (a) (demonstrating continuous layering). (f) similar to (e), an A-scope of relative received power (left) and the absolute value of the gradient (right) for a trace in panel (b) (characterising disrupted layering). For both (e) and (f), the black line depicts the whole trace whilst the blue is the section used for calculation of the ILCI.

**4 Results**

ILCI results derived from our 100-trace (2 km) and 1000-trace (20 km) windows reveal four macro-scale zones across the Lambert Glacier catchment (Fig. 3):

Zone 1: ILCI values are generally high (>0.45) in the upper catchment, before the onset of enhanced ice flow ( i.e. >15 ma$^{-1}$). This zone nominally represents an area of continuous englacial layering (e.g. Fig. 2a).

Zone 2: Very low ILCI values (<0.3) dominate the glacier onset zone, where ice currently transitions from low flow speeds to
faster streaming flow. These low ILCI values likely reflect buckled and disrupted ice structures (see Fig. 2b).

Zone 3: Variable ILCI values (0.25-0.5) are returned in Zone 3, within the lower catchment. Here, areas of high ILCI (>0.4) are transected by two narrow zones of low ILCI (<0.3) which coincide with channelised, fast glacier flow through deep subglacial valleys.

Zone 4: ILCI values are relatively low (<0.4) around Lake 90°E and Sovetskaya Lake, where there is a region of slow-flowing
thick ice (>3000 m).

**4.1 Englacial continuity, ice-flow and bed topography**

A spatial correspondence between ILCI returns and the current ice-flow regime at the ice surface is apparent across the Lambert Glacier catchment (Fig. 3a). In Zone 1, up-ice of the onset zone (Fig. 3a), no ice flow exceeds 10 ma$^{-1}$ across an area of 38,000 km². This slow-flowing inland ice is characterised by high ILCI values (>0.45), where similar slow-flow conditions throughout
the ice column help to maintain englacial stratigraphic continuity. Downstream of Zone 1 and into the Lambert Glacier onset zone (Zone 2), the ILCI signature evolves as surface ice-flow speeds increase (Fig. 3a). Low ILCI, associated with disrupted or absent layering (<0.3), begins to occur where ice-flow speeds pick up, to between ~10-15 ma$^1$, including through a topographically controlled tributary-like zone (marked with a T in Fig. 3) where ice-flow remains below <15 ma$^{-1}$. Through much of the rest of Zone 2, ice is channelled through substantial ice-flow trunk that connects the onset zone of Lambert Glacier
to the Amery Ice Shelf (Fig. 4). Fast-flowing ice in this trunk is associated with low ILCI (Fig. 3 (Zone 2)) and disrupted englacial layering (Fig. 4). However, down ice flow, through Zone 3, the ILCI is bimodal. Ice in the main trunk of Lambert Glacier displays the expected association of low ILCI values (<0.3) where ice-flow speeds are high (> 100 m a$^{-1}$). Elsewhere, however, and throughout Zone 3, ILCI is often >0.45. These ILCI values are typically associated with continuous englacial layering, which is surprising, given that surface ice velocities are generally quite fast flowing, and always in excess of 20 ma$^{-}$
$^1$. Zone 4 (Fig. 3) is beyond the main AGAP-N radar survey grid, and therefore characterised instead by six reconnaissance AGAP-N RES survey lines grid-south of Lambert Glacier. Although ice velocity is <10 ma$^1$ across the entirety of this zone, ILCI values are <0.4 contrasting the assumption that ice flowing slower will have a high ILCI return. Applying a lower threshold for "high" ILCI might highlight higher resolution changes in continuity in Zone 4, however, as we are primarily concerned here with the regional picture, this type of detailed analysis is beyond the scope of the present study.

The bed topography of the upper catchment (Zone 1) is dominated by the high relief of the subglacial Gamburtsev Subglacial Mountains, beneath Dome A (Morlighem et al., 2020) (Fig.s 1, 3a). When averaged over 1000 traces (Fig. 3b) the ILCI analysis does not appear to capture the impact of these subglacial mountains on englacial continuity. However, when ILCI is averaged over 100 traces (Fig. 3a), variations in ILCI across Zone 1 suggest an association between the steep subglacial bed slopes and disrupted or absent englacial layering above. Grid east of the upper catchment there is a region of lower bed relief with higher,

more continuous, ILCI values (>0.5). The high ILCI in this region correspond with parts of the catchment with ice thicknesses <2000 m (Fig. 1c). As ice enters the deep graben-like rift valley through which ice flow is currently focused, there is a spatial correspondence between increased ice velocity (i.e. >15 ma$^{-1}$), the deepening of the valley (~400 m asl to ~ -200 m asl) and low ILCI values (<0.3). Grid west of the onset, a tributary-like valley (marked with a T in Fig. 3) demonstrates low bed elevation (1,000 m b.s.l. at the deepest point), and low ILCI values (<0.3). However, ice velocity here is slower (<10 ma$^{-1}$),

which implies that the roughness of the bed topography is the likely cause of low reflector continuity. Downstream of the onset zone, in Zone 3 (Fig. 3b), the ILCI analysis shows a correspondence between the englacial architecture and bed topography. Adjacent to the central glacier trunk and rift valley, broad regions (~250 km wide) of flatter bed relief and large, relatively flat, subglacial platforms are associated with higher ILCI values (>0.45), reflecting continuous englacial layering throughout the ice column, and the layering is disrupted only where fast flow follows the deep geological rift and at the shear margin (Fig. 4).

Zone 4 contains a similar area of flatter bed relief (Fig. 3b) but, unlike Zone 3, this zone is characterised by lower ILCI values (<0.4). We note wave-like birefringence patterns in Figure 4, and in other radargrams within this paper (Figures 5 and 6). Analogous birefringence patterns in airborne radar data have been recorded and discussed elsewhere (e.g., Gerber et al., 2023; Young et al., 2021) and have been attributed to ice sheet bulk fabric anisotropy. In principle this birefringence effect has the potential to impact the derivation of ILCI values, because it may overprint the radar reflections, effectively introducing an

artefact to the data, and skewing the ILCI values. However, it is likely that the most prominent effect of this would be to overemphasise continuity in the results, which is not seen in our dataset. Here, the birefringence effect does not impact on high ILCI values associated with disrupted and buckled layers because the typical form of the birefringence pattern is substantively distinct from disrupted and buckled layers. A clear example of this is shown for the Lambert onset zone (Figure 4), where layer disruption and high ILCI values are readily apparent and do not correspond to variations in the visibility of the

birefringence.

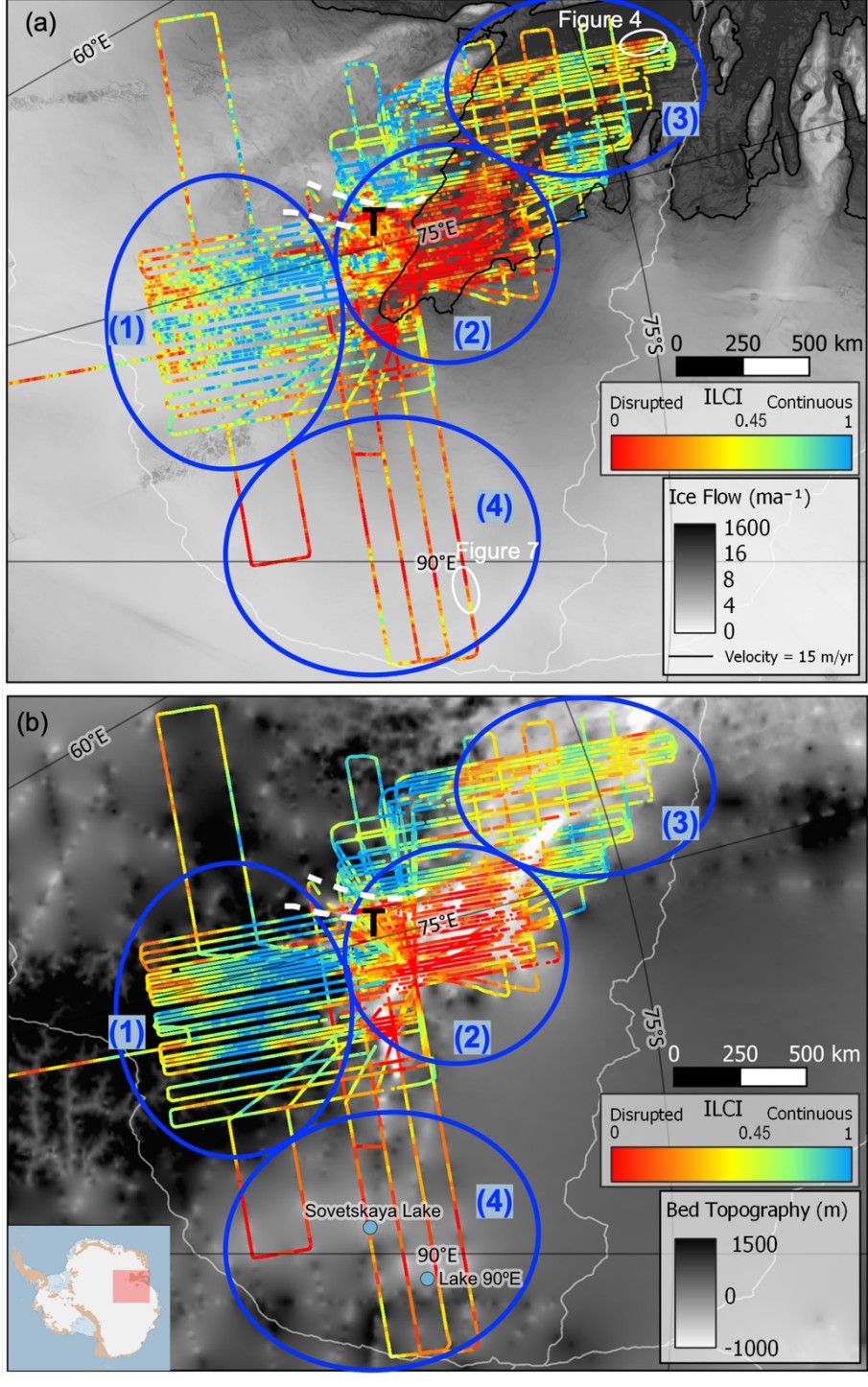

**Fig. 3. ILCI analysis demonstrating the four zones of ILCI returns in the Lambert Glacier. White line is the Lambert-Amery catchment boundary (Zwally et al., 2012a) (a) ILCI smoothed over 100 traces, overlying MEaSUREs ice-surface velocity**

measurements (Mouginot et al., 2019). A velocity contour where ice velocity >15 ma$^{-1}$ is marked with a black line. (b) ILCI analysis smoothed over 1000 traces, overlying BedMachine v2 bed topography (Morlighem et al., 2020). Blue circles depict the four dominant ILCI zones within [1] the upper catchment; [2] the onset zone; [3] downstream; and [4] grid south of the onset. A tributary feeding Lambert Glacier from grid north is denoted with a T and white dashed lines highlight the tributary. White circles on Fig. 3a mark the location of RES transects displayed in Fig.s 4 and 7. Sovetskaya Lake and Lake 90°E are labelled with a light blue circle in Fig 3b.

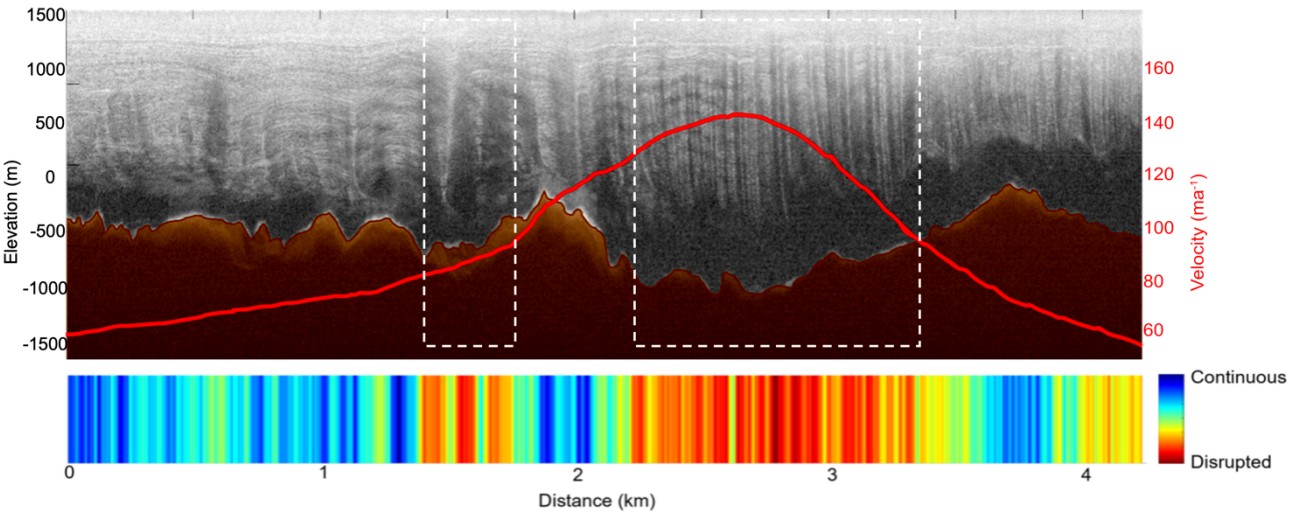

**Fig. 4. Ice velocity (red curve) (Mouginot et al., 2019), bedrock traced from the radargram (brown shaded area), and Internal Layering Continuity Index (ILCI) returns from RES line A29C. White boxes highlight disrupted englacial layering where ice flows through the central rift and at the margin of Lambert Glacier. The location of this transect is noted in Fig. 3a. Birefringence patterns visible in the radargram, especially between 2 and 2.5 km and between 0-1000 m elevation, are discussed at the end of Section 4.1.**

## 4.2 Englacial folding

The spatial correspondence between ice flow exceeding 15 ma$^{-1}$, converging ice flow into the Lambert Rift and the marked shift in ILCI values in Zone 2 (Fig. 3a) suggests that the onset of ice flow is important for the englacial structure of Lambert Glacier. Although ILCI values are relatively low in this zone (i.e. <0.4), radargrams show lateral continuity of englacial structures across multiple flight lines, where recognisable units or structures can be mapped from radargram to radargram down-ice (Fig. 5). Englacial layers at the onset of enhanced flow (>15 ma$^{-1}$) undulate in distinct trough and crest sequences, similar to folds observed elsewhere in Antarctica and Greenland (Bell et al., 1998; Conway et al., 2002; Ng et al., 2004; Holschuh et al., 2014; Franke et al., 2022a). The axes of these folds are subparallel to the current flow direction, with fold wavelength decreasing down flow from the onset zone, as ice flow converges. The amplitude of folding varies in the ice column and increases with depth. Eighteen distinct flow bands (defined by prominent englacial layer trough folds) have been identified within the region of ice-flow onset (Fig. 5). All flow bands propagate down-ice as the ice flows through the subglacial

rift valley beneath Lambert Glacier. Although the fold axial traces are typically vertical in orientation, the lowermost parts of some diverge from the vertical close to the bed (e.g. within 1 km above the bed). An example of this can be seen approximately 25-30 km along RES transect A46B-A46B' and A54B-A54B' (Fig. 5) where the fold axial traces deflect towards the centre of the valley. In each case, the deflection of the fold axes appears just above a zone of basal ice that is characterised by few or no internal reflections (i.e. an 'echo-free zone').

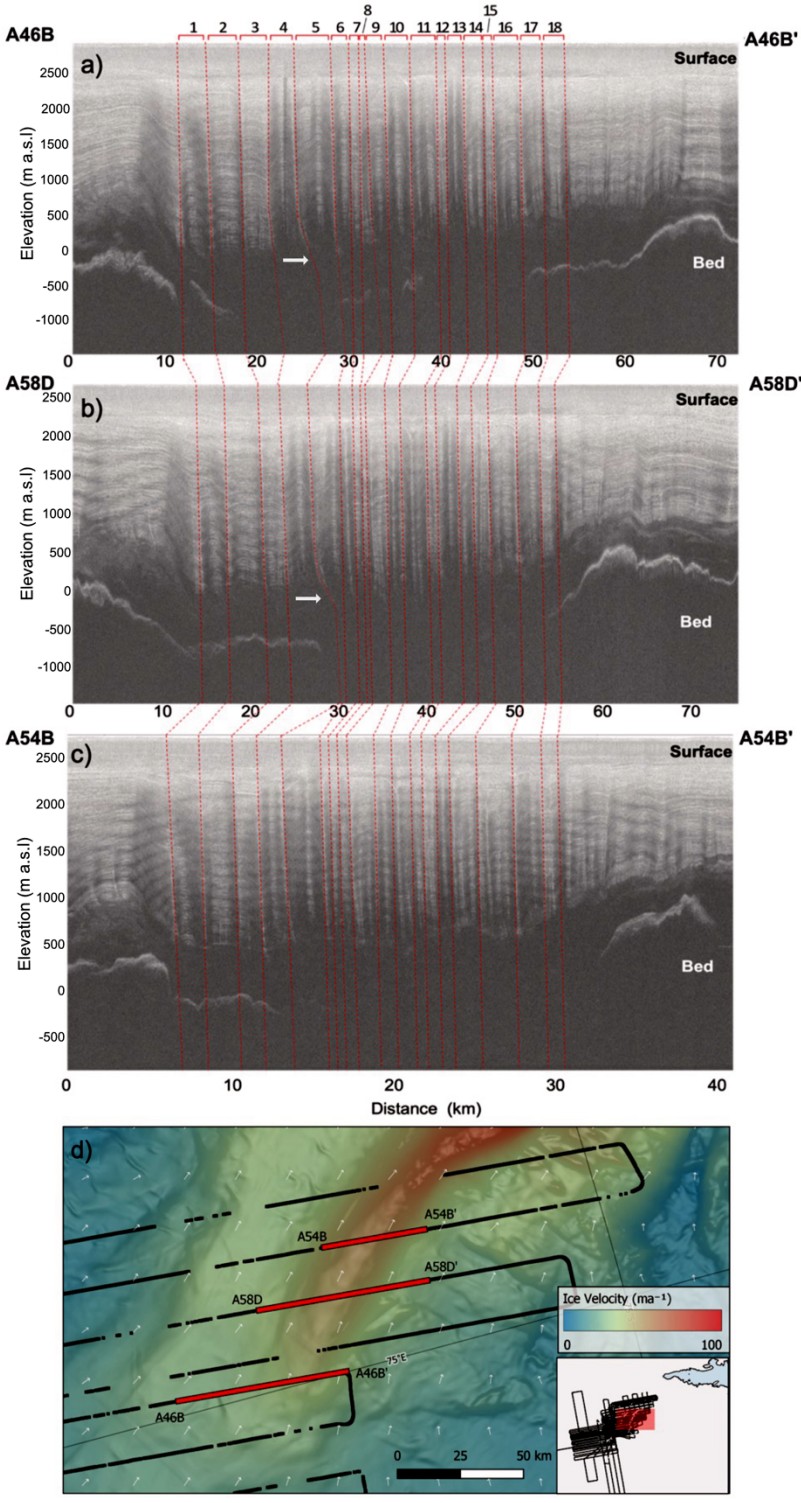

**Fig. 5. Radar profiles a) (A46B-A46B'), b) (A58D-A58D') and c) (A54B-A54B') showing englacial layering and the subglacial bed. Flow direction is from panels A to C. Eighteen flow bands, defined by a down-ice-warping in englacial layers, are mappable down-ice from panel A to panel C (dashed red lines). The white arrows in panels a and b show where the fold axial deflects. d) Current ice velocity and flow vectors (white arrows) overlying a raster derived from calculating the slope from the maximum horizontal gradient of ice velocity. This underlying raster has a grey-scale colour pallet and is histogram equalised with the colour scale saturated at 65 ma$^{-1}$ (Mouginot et al., 2019) of the onset of Lambert Glacier with flow direction from bottom left to top right; red lines denote profiles in (a), (b) and (c). Birefringence patterns visible in the radargrams (a,b and c) are discussed in Section 4.1.**

In addition to the short-wavelength folds observed at the ice-stream onset in Zone 2, large-scale englacial structures are visible in radargrams collected along the western margin of Lambert Glacier as ice flow converges towards its central trunk. A single large-scale fold (>360 km long, 5-10 km wide), characterised by a diagnostic, visually-traceable englacial structure, is apparent in the slower-flowing ice outside the Lambert Glacier tributary shear margin (Fig. 6). This englacial feature is visible in a series of 36 radargrams, through which it can be traced ~360 km down-ice of the onset zone (Fig. 6a). The limits of the survey grid restrict further tracing of this feature, but we are confident that the fold extends further upstream and downstream than the 360 km long extent we map here. As ice velocity increases down flow, from 15 ma$^{-1}$ to >100 ma$^{-1}$, the width of the fold reduces, and its amplitude increases. The fold axis runs parallel to ice flow in an area of accelerating and converging ice-flow (where velocity ranges from ~ 15 ma$^{-1}$ to ~ 50 ma$^{-1}$) (Fig. 6b), and is located immediately adjacent to the central rift valley of the enhanced flow zone of Lambert Glacier (Fig. 6c). Assuming that the fold formed and then advected down-ice with flow, we estimate that this englacial fold could have persisted for at least 10.5 ka (based on calculations of the average current ice velocity (Mouginot et al., 2019) and the distance between each fold on individual flightlines). It is likely that the fold extends further inland and we cannot visualise the exact location of the fold formation from existing data. We define the initiation point of the fold as the southernmost yellow dot (Figure 6a), corresponding to the southernmost radargram of the AGAP-N survey. In most radargrams, the fold is unrelated to the bed topography (i.e. the layers do not drape over a localised topographic high) and the fold axis is parallel to macro-scale ice flow. Similar scale folds, associated with convergent ice flow and the shear margins of outlet-glacier tributaries, have been observed in Greenland and Antarctica (Jacobel et al., 2000; Bons et al., 2016; Ross et al., 2020; Siegert et al., 2004). The amplitude of the folding (~200 m vertical relief) is greatest in the lower Lambert catchment (around Zone 3) and corresponds with pronounced convergent ice-surface flow stripes imaged on RADARSAT (Ely et al., 2016) that seem to initiate in response to accelerating ice flow at the onset zone (Fig. 6). One of these flow stripes corresponds with the fold axis of the traceable englacial fold, and extends the full ~360 km down-ice (Fig. 6). The surface flowline associated with the peak of the fold is undetectable in RADARSAT imagery in the upper catchment where ice velocity drops below ~10 ma$^{-1}$. However, the englacial fold extends significantly further up-ice of this point, existing further into the upper catchment than the overlying surface flow stripe with which it is associated.

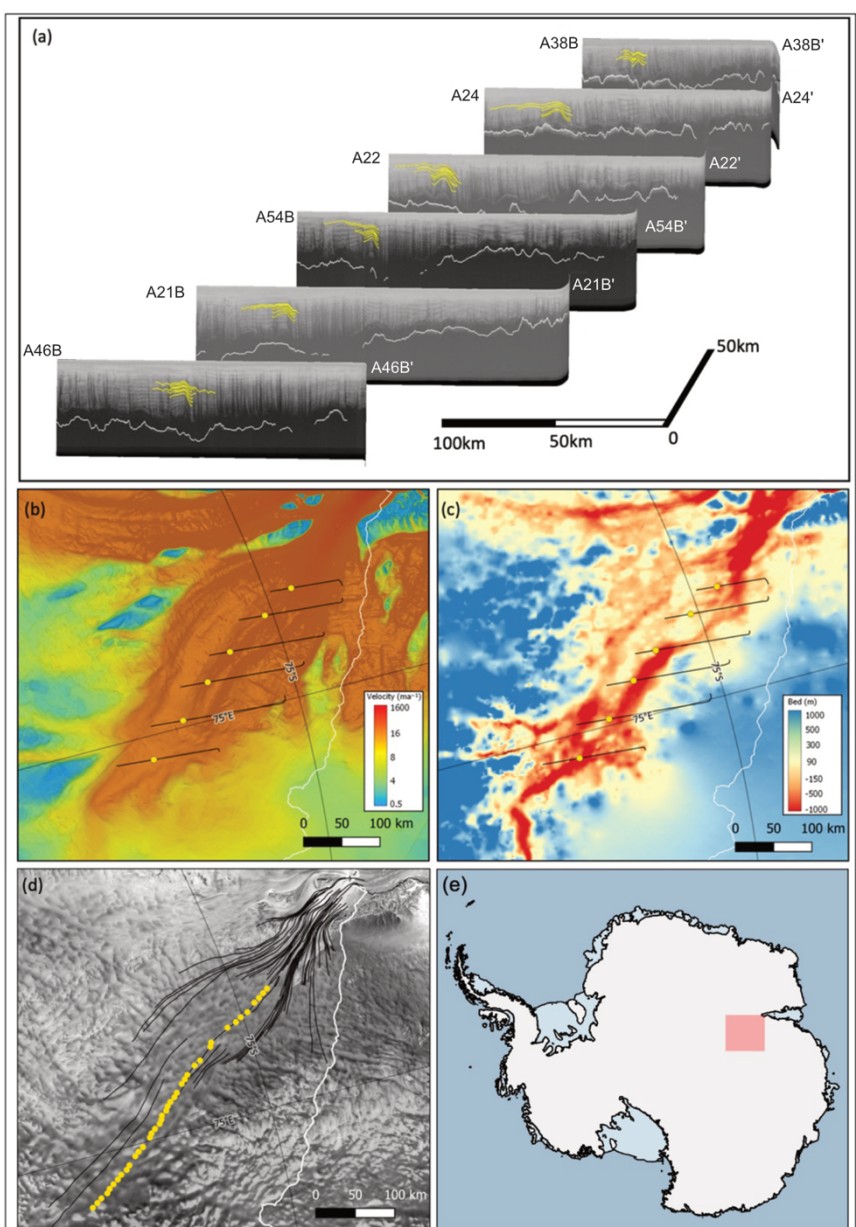

**Fig. 6. (a) Englacial-layer folds within a selection of radargrams from upstream transect A46B-A46B' to down-ice transect A38B-A38B'. Locations of lines A46B-A38B shown on b and c ; (b) Fold axis locations mapped with a yellow circle, along transects A46B to A38B, underlain by ice-flow speed overlying a raster of the maximum horizontal gradient (65) of ice velocity with a grey-scale colour pallet (Mouginot _et al._, 2019); (c) Fold axis locations mapped between lines A46B and A38B, underlain by bed elevation (Morlighem _et al._, 2020). (d) Location of surface flow stripes (Ely and Clark, 2016) (black lines) and peaks of the englacial fold axes (yellow dots) across the wider AGAP-N radar dataset, overlain on RAMP RADARSAT radar imagery of Lambert Glacier (Jezek 1999). In panels b-d, the white line denotes the boundary of the Lambert-Amery catchment.**

## 5 Discussion

### 5.1 Catchment-scale englacial architecture of Lambert Glacier

The overall pattern of ILCI across the Lambert catchment has a spatial correspondence to the bed topography. Ice flowing over the subglacial highlands (northern Gamburtsev Province) upstream of the Lambert Glacier onset zone (Zone 1) has low ILCI (<0.3) when averaged over 100 traces. This low ILCI in the upper ice-stream catchment (Zone 1) is not associated with current enhanced or fast ice flow (as the ice currently flows at <5 ma$^{-1}$) (Fig. 3), and there is no evidence of past enhanced/fast ice-flow features such as buckled layers at depth) in the radargrams (e.g. Winter et al., 2015). There are three likely causes for the low ILCI values in this zone, the first of which we favour. First, the low ILCI could be a consequence of thick warmer ice, causing attenuation of the radar signal, resulting in poor imaging of deeper englacial layers (Matsuoka et al., 2012). Weak reflections caused by low amplitude power returned from deep layers would impact the ILCI as the method involves assessing the variability of the internal layer reflection on individual traces (Karlsson et al., 2012). Second, it is possible that disruption of the layers giving rise to low ILCI is a result of power loss from dipping englacial reflectors, as seen elsewhere in Antarctica (Holschuh et al., 2014; Winter et al., 2015). However, although low ILCI (<0.3) is apparent in areas of steeply inclined subglacial topography in the upper Lambert catchment (Zone 1), the relationship between ILCI and basal topography is not as evident as that between low ILCI and ice thickness. Third, it is possible that thick ice, converging through topographically controlled bed topography and/or refreezing at the bed under pressure, is causing the formation of folds and basal ice packages (Bell et al., 2011; Wrona et al., 2018). These basal ice units would disrupt the ILCI result as they would not contain traceable reflections between traces. However, they are unlikely to have a significant impact on the ILCI, as the lower 20% of the ice thickness (where these basal ice packages are found) was removed prior to ILCI analysis.

In Lambert Glacier's onset zone, where enhanced flow (>15 ma$^{-1}$) initiates, the ice is channelised into the deep rift valley that constrains Lambert Glacier (Fig. 3b). Buckling of englacial layers is the most likely explanation for the low ILCI values returned in zones 2 and 3 (Fig. 3), as diagnosed for other areas of Antarctica where ice flow speeds up (e.g. Siegert et al., 2003; Bingham et al., 2007). Discontinuous and buckled ice layers dominate the full ice column here, as a result of relatively high current ice-flow speeds (~50 - 150 ma$^{-1}$) and/or ice converging into the Lambert Rift. Higher velocities throughout the ice column, caused in part by the presence and properties of basal temperate ice and associated basal sliding (Pittard et al., 2016), allow older ice in this part of the Lambert system (Zone 2 and 3) to be transported more rapidly towards the grounding line. Pittard et al. (2016) modelled the basal melt rate for the glacial system and found the average to be just 1.3 mm a$^{-1}$; however, they also note a large potential variability in the modelled melt, with a maximum modelled basal melt rate of >500 mm a$^{-1}$. We propose that variable basal sliding, in combination with converging ice flow and increased ice-flow speeds across Zones 2 and 3, are the primary cause of low ILCI values in the region (which qualitatively define visible internal layer buckles) (Fig. 5). Although this relation may also hold for ice in the main glacier trunk flowing through the Lambert Rift, ILCI returns here (>0.4) suggest more continuous englacial layering across the rest of lower Lambert Glacier, despite presently enhanced ice-

flow at the surface. This suggests that in the lower Lambert Glacier (i.e. zone 3), away from the main glacier trunk, englacial layers are more continuous than they were in ILCI zone 2 (around the onset zone) (Fig. 3a). The higher ILCI in the wider, lower catchment (i.e. outside the main trunk) may be attributed to: (i) a lack of significant subglacial topographic variation disrupting the ice; or (ii) colder, slow-flowing ice adjacent to the main trunk (Dawson et al., 2022). Although Payne and Baldwin (1999) found that colder, slow-flowing ice can control the location of major ice streams, Young et al. (2019) have stressed more recently that colder temperatures increase stiffness and can therefore reduce deformation of internal layers and hence induce less buckling and disruption.

Zone 4, the region grid south of the main Lambert Glacier channel (Fig. 3), demonstrates a breakdown of the assumption that low ILCI values and discontinuous layering occur as a result of changes in bed topography, converging ice and/or enhanced ice velocity. In this region, the bed is flatter than the upper catchment and the ice is slower flowing (i.e. <10 ma$^{-1}$) (Fig. 3), although the ice is considerably thicker (typically >3000 m thick) than across the majority of the catchment (typically <2000 m thick) (e.g. Fig. 7). Similar to the low ILCI region Zone 1 (Fig. 3), we suggest that the thick ice here could be increasing the attenuation of the radar energy, generating weaker reflections with implications for the ILCI analysis. Because 20% of the lower ice column is removed from the analysis, it is unlikely that basal ice units will impact the ILCI values. We therefore suggest that thicker ice causing weaker reflections is the most plausible cause of low ICLI values. However, another possibility is that the thicker, warmer ice in this area (Van Liefferinge and Pattyn, 2013; Dawson et al., 2022) could also reduce the amplitude of englacial layer undulations and mask continuous englacial layers. A broader point relevant to wider Antarctic studies beyond the Lambert region studied here, is that englacial layers in Zone 4 remain clearly traceable in the radar data despite generating low ILCI (Figure 7). This demonstrates that while ILCI typically acts as a useful first filter for assessing layer continuity (c.f. Frémand et al., 2022), prospects for tracing layers should not be dismissed across low ILCI zones.

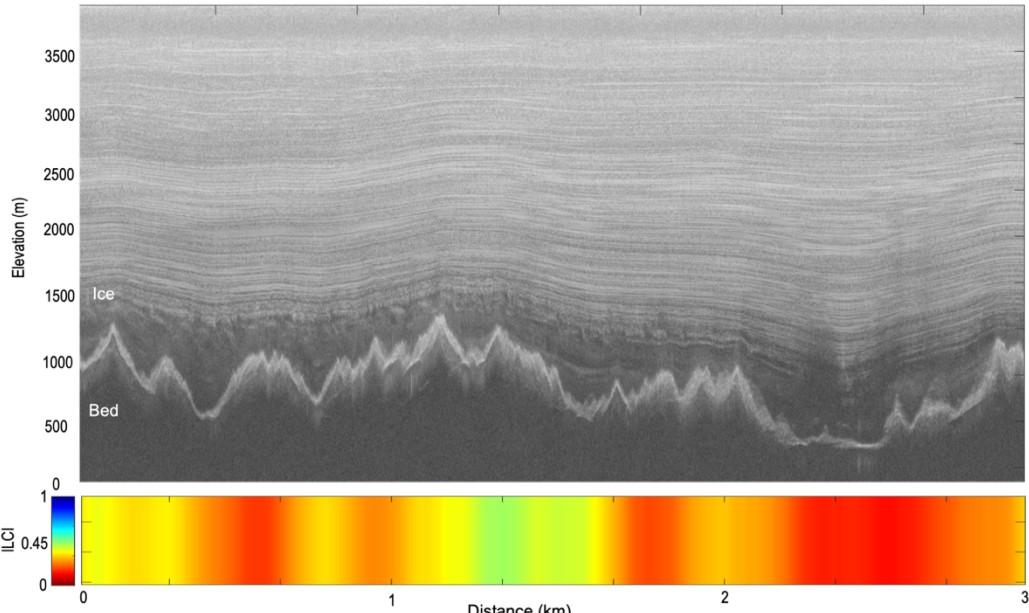

**Fig. 7. Example of RES transect (A33) exhibiting visible continuous layers, but a relatively "low" ILCI return. The mean ILCI return for the transect is 0.38. The location of this transect is detailed in Fig. 3a.**

## 5.2 The onset zone of Lambert Glacier

The disrupted englacial-layer geometry at the onset of Lambert Glacier (Zone 2, Fig. 3) is comparable to that observed elsewhere in Antarctica (e.g. the tributary onset zones of Pine Island Glacier (Karlsson et al., 2012)), where englacial layers are similarly folded and buckled. As we assume that these buckled layers are the product of differential ice motion caused by lateral shear stresses at the transition from slow to faster flow, and the ice convergence into the Lambert Rift (Siegert et al.,

2003), ILCI evidence of disrupted englacial layers up-ice of the onset zone suggests that there may not be an abrupt no-slip to sliding transition beneath Lambert Glacier. Instead, a more gradual shift to basal sliding, over a distance of approx. 200 km, appears to occur. An alternative view is that the onset zone could have moved over time. We explore these two ideas using pre-existing research from other Antarctic field sites. At Pine Island Glacier, ILCI analysis of englacial layering suggested that while some areas demonstrated a similar gradual transition to basal sliding, in other areas the transition between internal

deformation and basal sliding is very sharp (Karlsson et al., 2012). Karlsson et al. (2012) attributed this abrupt transition to a distinct qualitative difference between disturbed and absent layering at a tributary confluence. This setting is not particularly analogous to Lambert Glacier, where there is a broad onset zone, which is not influenced by multiple tributaries like Pine Island Glacier. In this respect, the onset zone of Lambert Glacier appears to be more analogous to that of Bindschadler Ice Stream, West Antarctica, where an analysis of longitudinal stress gradients has shown that a shift to basal sliding occurs as an

integrated response of bed and ice interactions upstream of the onset zone, rather than changes in basal characteristics at a set

location (Price et al., 2002). The initiation of flow bands at the onset of Lambert Glacier (Fig. 5) (Ely and Clark, 2016) is not coincident with highs in bed topography and therefore the flow bands are likely to have formed as a result of differential basal conditions causing high basal sliding at the location where ice is converging (and the resultant speed up of ice flow) (Ely and Clark, 2016; Wolovick et al., 2014). The correspondence of disrupted and sloping englacial layers and elevated basal shear

stress in the Lambert onset zone (Morlighem et al., 2013; Sergienko et al., 2014) supports the theory of an extended gradual transition from a frozen to temperate bed (i.e. from internal deformation to basal sliding), as suggested for Institute Ice Stream in West Antarctica (Bryant et al., 2019; Mantelli et al., 2019). In contrast, tributary 1 (marked with a T on Fig. 3) returns low ILCI returns when ice flows through the subglacial valley. Although these ILCI returns suggest disrupted ice as a result of converging flow and/or increased ice velocity, this is not observed at the present-day ice surface (Mouginot et al., 2019). The

disruption to the englacial layers suggested by the ILCI is possible evidence that tributary 1 previously experienced enhanced ice flow. This would have been the result of either previous enhanced ice flow in the tributary, or a migration of the onset region of Lambert Glacier, with ice flow from a more grid north direction. Similar tributary flow switching is thought to have occurred at Institute Ice Stream in West Antarctica (as evidenced by RES transect and ILCI analysis) (Winter et al., 2015). In summary, we posit that the onset zone of Lambert Glacier has a gradual transition from internal deformation to basal sliding,

but there is evidence of potential flow changes having occurred there in the past.

The trunk of Lambert Glacier is characterised by low-continuity ILCI values and buckled englacial layers which are first recorded at, or just upstream of, the ice-flow onset zone (Fig. 3). We suggest that buckled layer geometry in the glacier trunk, as it flows through the subglacial rift valley, forms because of changes to the internal-strain field upstream, as fast ice becomes

channelised and/or as large subglacial protuberances disrupt the strain field when ice flows across them. This is further evidenced by the presence of flow bands, consisting of troughs and crests in multiple radargrams (Fig. 5). Persistent flow bands and buckled folds across tens of km suggest a steady long-term direction of ice flow, as changes in flow direction would have altered the persistence and orientation of these channel buckles.

Basal sliding can cause spatial and temporal differences in ice-flow speeds, such as at the onset of Lambert Glacier, and this is often attributed to the presence of meltwater at the glacial bed. Subglacial hydrology has therefore been inferred to play an important role at the onset of faster-flowing ice or ice streams (Bell et al., 2007; Franke et al., 2020; Siegert and Bamber, 2000). There is limited evidence of subglacial lakes in the onset zone of Lambert Glacier (Zone 2, Fig. 3) (Livingstone et al., 2022). However, there are numerous subglacial lakes in the upper catchment of Lambert Glacier, beneath Domes A and B

(Fig. 1) (Livingstone et al., 2022) which could supply basal water to the onset zone, and two very large lakes (e.g. Lake 90°E (2000 km$^2$) and Lake Sovetskaya (1600 km$^2$)) are located grid south in the catchment (Bell et al., 2006).These large lakes are tectonically controlled and considered stable (Bell et al., 2006) so large-scale production of meltwater up-ice from these is unlikely to play an important role in the transition from internal deformation to sliding beneath Lambert Glacier.

## 5.3 Englacial structure of Lambert Glacier shear margin

Large-scale englacial folds are located just outside the Lambert shear margin and persist throughout the ice column, and over a distance of 360 km (Fig. 6). The consistent orientation of the fold axis of these features within multiple radar lines transecting the glacier suggests that no significant past changes in flow direction from the current ice-flow regime are recorded in this part of the ice sheet. This supports Glasser et al. (2015), who noted that ice parcels at the onset of Lambert Glacier and other glaciers in Antarctica such as Recovery, Pine Island and Byrd glaciers may have resided for ~ 2500 to 18,500 years. Likewise, the

topographic constraint of the rift valley within which the glacier flows imposes a fixed width to Lambert Glacier, restricting the shear margin from migrating easily in response to perturbations in ice flow from external forces (Grinsted et al., 2022). The large-scale englacial folding we observe at the shear margin is therefore likely to be a result of lateral convergence, driven by enhanced ice flow through the subglacial valley (Franke et al., 2022a). However, we note that the englacial architecture of Lambert Glacier appears to be sensitive to variations in basal thermal conditions; where it is likely that ice compression causing

the fold is exaggerated by the transition from cold/frozen conditions outside the shear margin to the warm/ thawed conditions in the central trunk. Furthermore, Hills et al. (2022) note that the temperature gradient across the shear margin is likely to be stronger across the shear margin than along it. This helps to account for the fold advection we record down flow, where we find a similar englacial architecture for hundreds of km, with minimal change or disruption to the fold regime.

We have found the large-scale englacial fold along Lambert Glacier's shear margin to be directly manifested on the ice-sheet surface as a flow stripe (Fig. 6d). This study provides supporting evidence demonstrating that non-topographic englacial features and local variations in flow regime can lead to the development of surface features such as flow stripes (Cooper et al., 2019; Glasser et al., 2015; Ross et al., 2020). The fold along the shear margin extends significantly further upstream than the corresponding surface flowline. This is consistent with long-term persistence of ice flow in the Lambert catchment and may

provide evidence that the onset of enhanced flow once occurred further up ice flow than at present. The surface expression of the englacial folding (i.e the flow stripe), tied to the subsurface evidence from the radar, allows us to characterise and infer large-scale flow patterns over time (Ely and Clark, 2016). The persistent occurrence and orientation of the englacial fold suggests that the ice-flow pattern of Lambert Glacier has not changed for a substantial period. Based on calculations of the average current ice velocity (Mouginot et al., 2019), and the distance between each fold on the individual flightline, the

englacial fold at the Lambert Glacier shear margin is likely to have persisted for over 10.5 ka (i.e. throughout the Holocene), consistent with modelled evidence (Gong et al., 2014) for long-term ice-flow stability of Lambert Glacier. Increased understanding of ice margins is vital for understanding the mechanics of glaciers and ice streams, and how they can change in the future. The englacial layers across the margin warp but remain continuous along Lambert Glacier, as is the case across Whillans Ice Stream (Van Der Veen et al., 2007) indicating the sensitivity of englacial layers to high shear strain.

## 5.4 Wider implications for large outlet glacier systems

This study has provided an extensive evaluation of the englacial architecture of a large outlet glacier system in East Antarctica. Assessing large outlet glacier/ice-stream systems, particularly in East Antarctica, is important for ice-sheet mass-balance and their contributions to sea-level rise (Stokes et al., 2022). Although extensive research has been undertaken on ice streams (Winsborrow et al., 2010; Gerber et al., 2023), the recent acceleration of outlet glaciers as a response to climatic or internal perturbations remains poorly constrained (Jordan et al., 2018; Miles et al., 2021). By investigating the englacial architecture of these systems, and their relationships to the bed and surface, we can obtain an increased understanding of how perturbations at and near the ice-stream margin can propagate up-ice (e.g. up to the onset zone), and impact the flow of the interior ice. Likewise, our knowledge of shear-margin formation and migration has been developed, as has been the case for the North East Greenland Ice Stream (Franke et al., 2022b; Grinsted et al., 2022; Holschuh et al., 2019). Here we have provided a baseline characterisation of the Lambert Glacier system and have analysed RES data to examine current and past ice flow. If similar studies were applied to other large outlet systems across Antarctica, we will be able to provide more and better, localised constraints for ice-flow models, which will improve our current understanding of ice flow and discharge in the past and for the present day, with clear benefits from each, for understanding the processes driving ice sheet change, and consequently global sea level projections.

## 6 Conclusions

This paper has interrogated and characterised englacial stratigraphy across the Lambert Glacier catchment, the largest glacier ice-shelf system in East Antarctica. Our results highlight four distinct zones of englacial architecture. Zone 1 returns high ILCI values (>0.45) in the upper catchment, where englacial stratigraphic continuity is maintained by slow-flowing ice conditions. Zone 2 comprises of very low ILCI values (<0.3) in the onset zone where ice accelerates from slow flow by internal deformation to more enhanced flow (defined here as >15 ma$^{-1}$), which generates buckles and discontinuity in the englacial stratigraphy. Zone 3 defines an area with variable ILCI values (0.25-0.5) within the lower catchment, where areas of traceable, continuous englacial layers are recorded, as well as regions of discontinuity or englacial layer absence, particularly when ice-flow speeds are high (> 100 m a$^{-1}$). Zone 4 contains relatively low ILCI values (<0.4) outside the main flow of the Lambert Glacier, around Lake 90ºE and Sovetskaya Lake, where ice is ~3000 m thick, but slow flowing.

Whilst our results within Zone 1 are to be expected, and are comparable to findings from similar upper-catchment ice flows in Antarctica, we note interesting findings in all other zones. The englacial stratigraphy in Zone 2 demonstrates a gradual (rather than abrupt) transition from internal deformation to basal sliding at the onset zone. Combined with the minimal deformation and amplitude change of traced flow band fold axes in the region, this finding helps to reveal long term ice flow stability within the main trunk of the Lamber Glacier. This long-term stability aligns with the persistence of a large-scale englacial fold along the shear margin of the glacier (which we traced for 360 km), where analysis of the fold axis and orientation provide little

evidence for migration of the shear margin. These discoveries represent a novel understanding of one of the largest glacial ice-shelf systems in Antarctica, where findings are unlike those found in many other areas of the continent, where englacial stratigraphic analysis has revealed more dynamic ice flow changes within onset zones, and along glacier shear margins.


We do however note some evidence of change within the catchment. In ILCI Zone 2, where slow-flowing ice feeds into the main trunk of Lamber Glacier from the north (point T on Fig. 3), we find disrupted and discontinuous englacial layers which do not correspond to changes in the glacial bed, or indeed current ice flow conditions at the glacier surface. We therefore suggest that the disturbed englacial layers here evidence previously enhanced ice flow, within a former tributary of the Lambert

Glacier. Whilst this finding will have some implication for former ice flux calculations, it is possibly more notable for simulations of future glaciological change, as the tributary could provide a conduit for flow again in the future, were ice conditions to change as a result of internal or external forcings (i.e. climate warming).

In Zone 3 more continuous englacial layering was again recorded, despite ice having moved through a region of fast ice flow,

within Zone 2, to get to Zone 3. Whilst changes in the bed, and a resultant relaxation in ice flow could account for this transition in englacial layering, we hypothesise that these changes in ILCI may also reflect changing thermal regimes, as ice flows through the catchment and transitions from fast, warm flow, to colder, slower flow. Whilst similar thermal changes may be at play in Zone 4, where in this case, warmer ice could be reducing the amplitude of englacial layer undulations and masking continuous englacial layering (resulting in the unexpectedly low ILCI values we record). We suggest that our ILCI readings

could also be impacted by RES attenuation in thick ice flows, which generate weaker reflections, despite any real change in the englacial architecture.

Our findings highlight the applicability of ILCI tracing over large outlet glacier systems, and the importance of characterising englacial architecture across a catchment, in order to more fully understand the past and present flow conditions of the Antarctic

Ice Sheet. The findings also highlight the limitations of using the ILCI method in isolation and we recommend that these limitations be taken into account when the ILCI method is applied in future. Our results constrain key ice dynamics and provide flow information which will improve estimates of future mass-balance and sea level rise estimates. This is critical for projections of future global change, especially as East Antarctica contains a significant potential contribution to sea level rise.

**Data availability**

All datasets used in this paper are freely available online. The ILCI values and the englacial reflectors along the shear margin are published alongside the manuscript at  https://doi.org/10.25405/data.ncl.23708511.v1. The UK Polar Airborne Geophysics Data Portal hosts SEGY files of RES data for the AGAP-N survey (https://www.bas.ac.uk/project/nagdp/) whilst the National Snow and Ice Data Centre contains BedMachine (Morlighem et al., 2020) data (containing bed elevation, bed topography and

ice thickness information): http://nsidc.org/data/nsidc-0756 as well as ice-surface velocity data from MEaSUREs (Mouginot

et al., 2019): https://nsidc.org/data/nsidc-0754/versions/1. We use the freely available Quantarctica data set (https://www.npolar.no/quantarctica/) to view and interrogate RADARSAT mosaic imagery in QGIS (https://www.qgis.org/en/site/).

**Author contribution.**

The study was conceived by R.J.S, N.R, R.G.B, T.A.J, K.W and S.L.C. R.J.S performed data processing and analysis with contributions specifically from F.N with the ILCI code. R.J.S interpreted the results with continuous input from N.R, R.G.B, T.AJ, K.W and S.L.C. R.J.S wrote the paper, with edits from N.R, R.G.B, T.A.J, K.W and S.L.C.

**Competing interests.**

The authors declare that they have no conflict of interest.

**Acknowledgments.**

We would like to thank the AGAP team for collecting the RES data across the Lambert Glacier system, and the British Antarctic Survey (BAS) for making the data freely available and accessible online. We are also grateful for the collection and continual development of freely available datasets and data interrogation platforms like BedMachine, MEaSUREs, Quantarctica and QGIS. This paper was inspired by the AntArchitecture Action Group of the Scientific Committee for Antarctic Research and the RINGS Action Group of the Scientific Committee for Antarctic Research. R.J.S. was supported by the National Environmental Research Council (NERC)-funded ONE Planet Doctoral Training Partnership (NE/S007512/1), hosted jointly by Newcastle and Northumbria Universities. F.N. acknowledges support from the Agencia Nacional de Investigación y Desarrollo (ANID) Programa Becas de Doctorado en el Extranjero, Beca Chile, for a doctoral scholarship.

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
