# Peer review of "Englacial Architecture of Lambert Glacier, East Antarctica"

_The Cryosphere, 2023_

## Referee Comment (RC1)

Review of: **Englacial Architecture of Lambert Glacier, East Antarctica** by Rebecca J. Sanderson and others.
* * *
The paper focuses on analysing englacial stratigraphy from radio-echo sounding data from the Lambert Glacier catchment in East Antarctica. For the analysis, an internal-layering continuity index (ILCI) is applied to classify the survey region into areas of high and low layer continuity. Four areas are discussed in detail and the ILCI and other features, such as englacial folds, are set in the glaciological context, e.g. with ice velocity and bed topography. On this basis, interpretations are made on the causes of folding or other reasons for layer discontinuity, as well as interpretations of the ice-dynamic stability of the glaciological system over the last thousands of years.

The paper is very well written, logically constructed and well structured. The many illustrations give a good overview of the radar stratigraphy in connection with important features such as ice velocity, topography and features on the ice surface. As I understand it, this is the first major analysis of the radar stratigraphy of this area where a very large data set is covered. Due to the large amount of data, it makes sense to use an analysis tool like the layer continuity index, especially when it comes to getting a first overview of this region.

In my opinion, however, some of the interpretation of the results of the ILCI values, in particular, what they mean and how the ILCI values are influenced by the features in the radargrams, should be improved and complemented. In addition, I am not yet convinced regarding some of the interpretations and statements being made, especially regarding the development of folds. I also have a different point of view in a few passages in the text as well as questions for clarification.

I am aware that the analysis and interpretation of such data are complex and take time, simply because of the amount of data and number of different features it contains. The authors have a good job on focusing on certain aspects and the paper makes an important contribution, which is highly relevant to the glaciological community and for our understanding of the Antarctic ice sheet. Therefore, I would be very happy to see this paper published, even though I think that a few things should be presented or added in more detail. I hope that my comments and questions will help to clarify a few additional things and still unclear issues (which at least are not 100% clear to me). In the following, I will address a few main points as well as specific points in the text.
* * *
**Main points**

**1. Factors influencing the ILCI**
The ILCI is supposed to be a measure of the continuity of the internal layers and it would be interested to know if the following things were taken into account for the evaluation of the index:

- Does the orientation of the radar lines, e.g., with respect to ice flow direction or with respect to the fold structures in Zone 2 affect the ILCI values? If yes, how, and maybe one sentence could be added to section 3.2 in that regard.
- In the radargrams in Figure 4, 5a,b,c (especially in this one) and 6a a strong birefringence signal is visible. It would be interesting to know how the birefringence affects the ILCI and whether this has been considered in the calculation. My assumption is that if the birefringence patterns are aligned horizontally, they probably don't have much effect, but if they are tilted, they have an effect depending on the window size. If the birefringence affects the continuity index, then the interpretation of the results becomes even more complicated. Depending on how the fabric is pronounced in the horizontal plane in these regions and how the window size is chosen, the interpretation of the results becomes more complex, but also contains more information.
  The presence of this pattern should be mentioned in general (see Young et al., 2021; Gerber et al., 2022) and the potential effect on the ILCI should be included in 3.2 and the discussion, i.e. whether and how birefringence affects the continuity index and what the birefringence signal could potentially mean in terms of ice dynamics.

Young, T. J., Schroeder, D. M., Jordan, T. M., Christoffersen, P., Tulaczyk, S. M., Culberg, R., & Bienert, N. L. (2021). Inferring ice fabric from birefringence loss in airborne radargrams: Application to the eastern shear margin of Thwaites Glacier, West Antarctica. Journal of Geophysical Research: Earth Surface, 126, e2020JF006023. https://doi.org/10.1029/2020JF006023

Gerber, T., Lilien, D., Rathmann, N., Franke, S., Young, T. J., Valero-Delgado, F., Ershadi, R., Drews, R., Zeising, O., Humbert, A., Stoll, N., Weikusat, I., Grinsted, A., Hvidberg, C., Jansen, D., Miller, H., Helm, V., Steinhage, D., O'Neill, C., Gogineni, P., Paden, J., Dahl-Jensen, D., and Eisen, O.: Crystal fabric anisotropy causes directional hardening of the Northeast Greenland Ice Stream, Preprint (Version 1), Research Square, https://doi.org/10.21203/rs.3.rs-1812870/v1, 2022

**2. Linking ILCI values with ice flow velocity**
In the paper, the authors often connect low ILCI (i.e., disrupted layering) and high flow velocity. This is one of the statements repeated very often in the paper. However, my impression from the results of the analyses is that other specific correlations are the much more useful background information related to the disturbed stratigraphy and these should be brought to the foreground. It is true that, especially in Zone 2, the low ILCI is associated with higher flow velocity (and acceleration). However, I don't think that just an increase in flow velocity can disrupt internal layering, because it would only lead to along-flow dilatation, which should have no other effect on the layering than thinning. My feeling is that horizontal shortening in the ice that starts in this region of convergent flow and acceleration, that leads to englacial folding, which in turn leads to backscatter loss at the steep internal layers. This leads to a low ILCI, but at this point the ILCI itself is no longer relevant but rather the fold geometry, which allows further analyses and interpretation. In my opinion, the statement that

a higher flow velocity (magnitude) is related to disrupted layering is too simplistic. It coincides in this Zone but is not necessarily related to each other.

**3. The origin of flow bands**

The paper suggests that *so-called* flow bands are formed by variable basal sliding, and much of the discussion focuses on inferring basal shear stress in the region and comparing it with results from other studies of basal shear stress in other Antarctic ice streams. I am not yet convinced that the "flow bands", which to me are open cylindrical folds extending parallel to the ice flow, are caused by changes in basal shear stress. I have the impression that it is more of an assumption than a solid interpretation. The references to the literature in this context were not fully informative enough for me as to how the folds we see there are formed by variations in shear stress at the base of the ice. A long part of the discussion is then based on this interpretation, which I find problematic if this interpretation is not supported by more arguments.

My suggestion would be, on the one hand (if the authors keep this explanation as the most likely explanation for the formation of these folds), to explain in much more detail how differences in friction at the base produce exactly these folds (with fold axes parallel to ice flow). More references would also be helpful here, especially clearly referencing the mechanism and how it relates to the folds we see.

Secondly, an alternative explanation that the folds we see in Zone 2 in the graben System are caused by convergent flow leading to horizontal shortening (Bons et al., 2016) should be included. My impression is that the graben system itself is causing the convergent flow, as it is the only way for the ice to flow towards the coast. This mechanism would be independent of the basal properties and would fit particularly well with the alignment of the fold axes. In my opinion, it would be helpful to show flow lines in combination with the ILCI values and the fold axes (here called flow bands). The acceleration of the ice flow would compensate for the mass gain due to the convergence and only lead to the dilatation of the folds along flow, which should have no influence on the folds shape and only reduces the amplitudes. For me, this hypothesis is even more plausible than the theory listed in the paper. In any case, arguments should be listed that speak for or against the respective theory.

Bons, P. D., Jansen, D., Mundel, F., Bauer, C. C., Binder, T., Eisen, O., Jessell, M. W., Llorens, M.-G., Steinbach, F., and Steinhage, D.: Converging flow and anisotropy cause large-scale folding in Greenland's ice sheet, Nature communications, 7, 1-6, https://doi.org/10.1038/ncomms11427, 2016
* * *
**Specific points**

**L33-35:** I am not sure if the statement that the ILCI values are *"revealing the transition from internal-deformation-controlled to basal-sliding-dominated ice flow"* should be made (see further comments).

**L35:** *"which align"* → what does align, the fold axes of the folds?

**L93:** What is the *"(65)"* referring to?

**L98:** *"The RES data used for this study were acquired in 2007-2009 over the northern region…"* → over the northern region of the RES grid?

**L98-101:** What is the vertical and horizontal resolution of the RES data?

**L113 (Section 3.2):** I think it would be good to mention that the overall quality of the radar data and the processing level have an effect on the continuity index as well as "other" potential causes, such as surface clutter (which is not the case here) or as in this case potentially the superimposed birefringence signal.
Another thing that could be added is a short statement about the benefit of this type of analysis in contrast to other more time-consuming methods, such as tracing many IRHs. For me, the advantage of using an ILCI would be that certain information is available for a large data set, which gives first clues for the overall nature of the internal structure and where to apply further, more particular analyses.

**L122:** add a space *"tracks(Fig. 2)"*.

**L166:** To me, this section highlights that it is probably not an optimal idea to focus on linking the ILCI values to the magnitude of ice flow. First, because Zone 4 would be an exception (which, to be fair, is discussed later in the discussion) and second, because not the magnitude, but the pattern of flow seems to be the controlling factor in combination with the bed topography, the quality of the radar data and superimposed features, such as the birefringence signal.
What I think would be extremely helpful is to plot flow lines or some other marker for horizontal shortening (90° to ice flow), in particular at the onset of Zone 2, because it looks like the ILCI is rather a signature for convergent flow (which I would expect at the trough onset) creating buckled folds and where buckled folds are transported downstream.

**L188:** *"zone 1"* → Zone 1

**L200-201:** The ILCI values for Zone 4 are really surprising and confusing to me. When I have a look at Figure 7, which is located in this zone, the stratigraphy looks very continuous but the values are extremely low. Hence, there is something in this data that falsely gives low ILCI values. At this point, I would not conclude that the radar stratigraphy in this region suggests *"disrupted flow"*. If so, what would actually be *"disruptive flow "*?

**L216-217:** *"The spatial correspondence between ice flow exceeding 15 ma$^{-1}$ and the marked shift in ILCI values in Zone 2 (Fig. 3a) suggests that the onset of ice flow is important for the englacial structure of Lambert Glacier."*
→ I agree, but the question is what causes the change in ILCI (which is caused in this zone by the layer buckling, correct?). For me, it looks like this is also the region where ice flow would converge (maybe this could be checked), regardless of the change in ice flow velocity.

**L222:** *"fold width"* → maybe better "wavelength".

**L223:** *"The amplitude of folding increases with ice thickness".* → What does this mean, that larger folds are found where the ice is thicker in general or that the fold amplitude varies in the ice column and increases with depth?

**L225-229:** I think the structures you are referring to are not the fold axes but the axial traces (the line that connects the hinges of the synclines or anticlines in your radargrams vertically), or if combined to a plane, the axial planes: see https://cdn.eduncle.com/library/scoop-files/2022/7/can_image_1657899700300.jpg).
The fold axis would be oriented "horizontally" by combining the hinge points of the same anticline or syncline from one radargram to the next one, hence more or less parallel to the direction of ice flow.

**L233:** Here, it says *"fold bands"*, should it be "flow bands"? If not, what are fold bands?

**L236:** *"high-frequency"* → maybe better "short-wavelength"?

**L243:** *"The fold runs parallel to ice flow"* → Do you mean "The fold axis runs parallel to ice flow"?

**L243-244:** *"[...] in an area of accelerating and converging ice-flow velocities (~ 15 ma-1 to ~ 50 ma-1)."* → What are "converging ice-flow velocities"? What I think is meant is that ice flow is accelerating (from 15 to 50 ma$^{-1}$), and at the same time, ice flow is convergent. So, ice flow is indeed convergent there, which would support the fold formation hypothesis from Bons et al. (2016).
I'm not saying that it should be done, but something that would be very interesting to see is how much horizontal shortening (90° to ice flow) along one of the fold axes (possibly the central one) is actually happening and if that fits to the change in wavelength of the folds along ice flow.

**L245-247:** *"Assuming that the fold formed and then advected down-ice with flow, we estimate that this englacial fold could have persisted for at least 10.5 ka (based on calculations of the average current ice velocity (Mouginot et al., 2019) and the distance between each fold on individual flightlines)."*
→ This is an excellent approach to investigate the stability of the system. It would be interesting to specify (in one of the maps) where you define the starting point of fold formation. Would it just be the location where in the radargrams the fold appears for the first time, hence the southernmost of the yellow dots in Figure 6a?

**L250:** Maybe also add Siegert et al. (2004) here.
Siegert, M. J. et al. Ice flow direction change in interior West Antarctica. Science 305, 1948–1951 (2004)

**L267-269:** Could you specify which of the zones you are referring to?

**L269-270:** *"This low ILCI in the upper ice-stream catchment is not associated with current enhanced or fast ice flow".* → If this refers to Zone 4, it says something different than in the statement in L200-201 (disruptive flow).

**L269-271:** Is it possible to refer to a Figure to see which region and radargrams are referred to?

**275-277:** *"Second, it is possible that disruption of the layers giving rise to low ILCI is a result of power loss from dipping englacial reflectors, as seen elsewhere in Antarctica (Holschuh et al., 2014; Winter et al., 2015)."* → Has this been checked, or is it an assumption?

**279-283:** The last of the three possibilities (like all three actually) have been listed here as potential reasons. Is it possible to look at the radar data to determine which of the three causes the low ILCI? Are the ice packets, as in Bell et al., 2011 to be found here or not? Here again I see the problem that we do not learn much from the ILCI values per se. If something is to be learned about the englacial architecture, it should be checked or excluded what is causing the low values.

**L293-295:** *"We propose that variable basal sliding, and therefore variable ice-flow speeds across zones 2 and 3, are the primary reason for high ILCI values in the region (which qualitatively define visible internal layer buckles) (Fig. 5)."*
→ Shouldn't it be rater "low" values instead of "high" in these zones?
I don't fully understand how variable basal sliding in these zones would create the observed folds. If the vertical velocity profile changes along flow, you may end up with something like the moving patches of high and low resistance as proposed by Wolovick et al., 2014? This would create folds (under particular conditions), which would, however trend (the fold axis) 90° to ice flow. But the buckled folds observed here trend parallel (the fold axis) to ice flow. Also, it is proposed later that the change in basal shear strength is rather gradual than abrupt.
If this interpretation on how the folds in Zone 2 are formed is the only one presented here, there is more explanation needed as well as references on how variable basal sliding would create the observed folds. The explanation that I would add as an equally well possibility is that the folds are created via horizontal shortening due to the flow convergence (Bons et al., 2016) when the ice is flowing into the main trough. This would match the orientation of the fold axis and would be independent of processes at the ice base.

Wolovick, M. J., T. T. Creyts, W. R. Buck, and R. E. Bell (2014), Traveling slippery patches produce thickness-scale folds in ice sheets, Geophys. Res. Lett., 41, 8895–8901, doi:10.1002/2014GL062248.

**L305-312:** Maybe I need clarification, but in the first paragraph of Section 5.1 you proposed three possibilities for the low ILCI values (assuming that in section 5.1 you were referring to this zone). But here, only the attenuation is mentioned. How does this fit together?

**L319-320:** *"As we assume that these buckled layers are the product of lateral shear stresses at the transition from slow and fast flow (Siegert et al., 2003a), [...]"*

→ I had a look at Siegert et al., 2003 and tried to understand the argumentation chain: onset of fast flow -> variable lateral shear stresses --> leads to layer buckling as observed here. However, Siegert et al., 2003 write:
"It should be noted that no attempt is made to quantify the degree of layer buckling or determine the mechanics responsible for "buckled" internal layering. It is likely that buckled layers occur as a consequence of high longitudinal stresses within regions of enhanced ice

flow, and lateral shear stresses at the transition of fast- and slow-flowing parts of the ice sheet (Jacobel and others, 1993). The assumption made in this paper is that these stresses occur as a consequence of enhanced ice flow, and that internal layers will become more buckled the longer such stresses are applied."

Jacobel and others, 1993 discuss a lot of possibilities for their buckled folds, but I have the feeling that at the end there is no clear conclusion that basal shear stresses are responsible for the folds they observe. They also state: " Certainly, the shorter wavelength of the folds on either side of the bedrock rise is suggestive of greater lateral compression there."

→ which points towards the horizontal shortening theory.

I did not investigate the origin of the basal shear stress theory further and I had the impression that variations in basal shear stress is rather an assumption and one of many possible explanations. If there is literature that allows a clear connection to be drawn between the folds observed here and the basal shear stress approach, it would be good to discuss it here.

**L333-334:** *"[...] therefore the flow bands are likely to have formed as a result of differential basal conditions causing high basal sliding (and resultant ice flow speed up) [...]"*

→ My comment here is a repetition of previous comments that it is unclear to me how the change in basal properties leads to the folds in Zone 2.

**L421-422:** *"The englacial stratigraphy in Zone 2 demonstrates a gradual (rather than abrupt) transition from internal deformation to basal sliding at the onset zone."*

Again, a repetition of previous comments that it is not clear to me how the change in basal properties leads to the folds in Zone 2 and I am also not convinced that the folds in Zone 2 demonstrate this transition. If so, the relationships should be explained precisely and more clearly.

**L447-449:** It has already been partly addressed in previous paragraphs, but for me, a conclusion of this study is that the ILCI values alone are not very meaningful. The interesting thing is what they say about radar stratigraphy when looking at low and high continuity zones. Low values can indicate folded layers, but also low resolution of the radar system, poor visibility of the layers due to high attenuation or being influenced by signals in the radar data, which are not just internal layers.
* * *
I would like to thank the authors again for a very interesting article. My many (partly repetitive) questions and comments should not be perceived too critically, but rather reflect my interest in the topic and the results.

Steven Franke
March 10th, 2023

---

## Author Response (AR1)

Rebecca Sanderson
School of Geography, Politics and Sociology
Newcastle University
NE1 7RU

27th July 2023

**Re: [Paper ID # tc-2023-13] Englacial Architecture of Lambert Glacier, East Antarctica by Rebecca J. Sanderson et al.**

Dear Dr Elisa Mantelli, dear Reviewers, dear TC readers,

We would like to thank both reviewers for their insightful and constructive reviews of our manuscript, as well as yourself and the editorial team for handling the review process.

We are very pleased to see that both reviewers recognised the importance of our results and how these were presented in our manuscript. Both reviewers have provided us with some important comments, the incorporation of which into our revised version we hope will improve the quality of our manuscript.

In the following response letter, we begin by addressing the comments from Reviewer 1, followed by those made by Reviewer 2. We have formatted the comments of each reviewer in blue, and our responses in black below each comment. We have included the manuscript changes in red with reference to the revised manuscript lines.

We look forward to hearing your decision and would be happy to address any queries you might have.

With best wishes,

Rebecca Sanderson (on behalf of all co-authors)

Reviewer 1

*The paper focuses on analysing englacial stratigraphy from radio-echo sounding data from the Lambert Glacier catchment in East Antarctica. For the analysis, an internal-layering continuity index (ILCI) is applied to classify the survey region into areas of high and low layer continuity. Four areas are discussed in detail and the ILCI and other features, such as englacial folds, are set in the glaciological context, e.g. with ice velocity and bed topography. On this basis, interpretations are made on the causes of folding or other reasons for layer discontinuity, as well as interpretations of the ice-dynamic stability of the glaciological system over the last thousands of years.*

*The paper is very well written, logically constructed and well structured. The many illustrations give a good overview of the radar stratigraphy in connection with important features such as ice velocity, topography and features on the ice surface. As I understand it, this is the first major analysis of the radar stratigraphy of this area where a very large data set is covered. Due to the large amount of data, it makes sense to use an analysis tool like the layer continuity index, especially when it comes to getting a first overview of this region.*

*In my opinion, however, some of the interpretation of the results of the ILCI values, in particular, what they mean and how the ILCI values are influenced by the features in the radargrams, should be improved and complemented. In addition, I am not yet convinced regarding some of the interpretations and statements being made, especially regarding the development of folds. I also have a different point of view in a few passages in the text as well as questions for clarification.*

*I am aware that the analysis and interpretation of such data are complex and take time, simply because of the amount of data and number of different features it contains. The authors have a good job on focusing on certain aspects and the paper makes an important contribution, which is highly relevant to the glaciological community and for our understanding of the Antarctic ice sheet. Therefore, I would be very happy to see this paper published, even though I think that a few things should be presented or added in more detail. I hope that my comments and questions will help to clarify a few additional things and still unclear issues (which at least are not 100% clear to me). In the following, I will address a few main points as well as specific points in the text.*

We would like to thank Dr Steven Franke for his insightful comments which have helped us to improve the manuscript. We provide responses to each of the major and minor comments raised and how we have addressed them for the revised version of the manuscript below.

**Main points**

**1. Factors influencing the ILCI**

*The ILCI is supposed to be a measure of the continuity of the internal layers and it would be interested to know if the following things were taken into account for the evaluation of the index:*

*● Does the orientation of the radar lines, e.g., with respect to ice flow direction or with respect to the fold structures in Zone 2 affect the ILCI values? If yes, how, and maybe one sentence could be added to section 3.2 in that regard.*

Thank you for this point. Yes, the orientation of the radar lines with respect to ice flow and the flow structures can impact ILCI The most well-known issue from previous studies is that in faster-flowing ice regions where ice can be anisotropically buckled, the ILCI will capture the buckling in most directions but may record high ILCI (continuous layering) if the radar line being interrogated follows the flowline. This phenomenon has been discussed in detail by Karlsson et al. (2012) and Bingham et al. (2015), who both draw upon the 3-D depiction of englacial layering along buckled regions of Ng and Conway (2004) [all these papers are already cited in our manuscript]. In the updated manuscript

we have added a new sentence to acknowledge this issue but clarifying that we do not consider our ILCI results to be adversely impacted for this study because the vast majority of the radar lines do not parallel the ice flow or englacial fold axes.

Revised Line 133 – 137:

[…] of englacial layering. This disruption is understood to be the result of ice currently, or previously, experiencing significant deformation as it flows through an ice-stream onset or shear zone, or around/over significant subglacial protuberances (Bingham et al., 2015; Winter et al., 2015; Winter et al., 2016). Previous applications of ILCI have noted that buckling/folding might be missed if the radar lines parallel the fold axes (see depiction in Ng and Conway, 2004), which is typically where the radar lines have been gathered along or close to the ice-flow direction (Karlsson et al., 2012; Bingham et al., 2015); However in practice very few airborne radar lines in Antarctica (including those analysed in this study) have been acquired sufficiently close to the alignment of ice-flow for this effect to dominate ILCI interpretations. ILCI therefore provides a useful mechanism for reconnoitring conditions across large regions of an ice sheet, to characterise broad ice-flow conditions and guide more detailed investigations in regions of interest (c.f., Fremand et al., 2022). It is in this spirit that we apply the ILCI technique in this study. […]

● In the radargrams in Figure 4, 5a,b,c (especially in this one) and 6a a strong birefringence signal is visible. It would be interesting to know how the birefringence affects the ILCI and whether this has been considered in the calculation. My assumption is that if the birefringence patterns are aligned horizontally, they probably don't have much effect, but if they are tilted, they have an effect depending on the window size. If the birefringence affects the continuity index, then the interpretation of the results becomes even more complicated. Depending on how the fabric is pronounced in the horizontal plane in these regions and how the window size is chosen, the interpretation of the results becomes more complex, but also contains more information.

The presence of this pattern should be mentioned in general (see Young et al., 2021; Gerber et al., 2022) and the potential effect on the ILCI should be included in 3.2 and the discussion, i.e. whether and how birefringence affects the continuity index and what the birefringence signal could potentially mean in terms of ice dynamics.

Young, T. J., Schroeder, D. M., Jordan, T. M., Christoffersen, P., Tulaczyk, S. M., Culberg, R., & Bienert, N. L. (2021). Inferring ice fabric from birefringence loss in airborne radargrams: Application to the eastern shear margin of Thwaites Glacier, West Antarctica. Journal of Geophysical Research: Earth Surface, 126, e2020JF006023. https://doi.org/10.1029/2020JF006023

Gerber, T., Lilien, D., Rathmann, N., Franke, S., Young, T. J., Valero-Delgado, F., Ershadi, R., Drews, R., Zeising, O., Humbert, A., Stoll, N., Weikusat, I., Grinsted, A., Hvidberg, C., Jansen, D., Miller, H., Helm, V., Steinhage, D., O'Neill, C., Gogineni, P., Paden, J., Dahl-Jensen, D., and Eisen, O.: Crystal fabric anisotropy causes directional hardening of the Northeast Greenland Ice Stream, Preprint (Version 1), Research Square, https://doi.org/10.21203/rs.3.rs-1812870/v1, 2022

We are grateful to the reviewer for pointing out this interesting observation and citing the two relevant studies. The wave-like birefringence patterns observed in Figures 4, 5 and 6 do not represent internal layers and highlight the preferential orientation of the ice fabric as found in other parts of the ice sheet (Young et al., 2021; Besson et al.,2012).

We do not expect the observed birefringence to affect the interpretations as the ILCI method considers the reflected power on a trace-by-trace basis as a proxy for layer continuity. The ILCI assesses the reflective power gradient of the radar data, so reduction/amplification in overall power across the birefringent bands should be less of an issue as the layers should show up as high gradient areas. Indeed, our expectation that the observed birefringence does not influence the ILCI is borne out by our results: for example, Figure 4 demonstrates that although the birefringence effect is most visible from 2 to 2.5 km and between 0-1000 m elevation, this has little correspondence with the ILCI results (which point towards strong layer disruption).

Hypothetically birefringence bands could make the ILCI method overemphasise continuity when little or none is present, but this will depend on the frequency and amplitude of the birefringence bands. ILCI values are the mean of the absolute value of the derivative, so if the bands change gradually they in effect introduce an additional low gradient to the dataset, which would have minimal impact on the ILCI values. Considering the wider survey, evidence of birefringence 'banding' is largely restricted to areas of buckling and disruption at the onset and along the Lambert channel. We therefore argue that the overarching conclusions of the manuscript are not impacted by the birefringence effect.

*Besson, D., Doolin, N., Stockham, M., & Kravchenko, I. (2012). Radio-frequency probes of Antarctic ice birefringence at South Pole vs. East Antarctica; evidence for a changing ice fabric. The Cryosphere Discussions, 6(6), 4695-4731*

We have acknowledge these issues in the revised manuscript:

- In the revised caption to Fig. 4, we add the sentence:

"Birefringence patterns visible in the radargram, especially between 2 and 2.5 km and between 0-1000 m elevation, are discussed at the end of Section 4.1."

- In Section 4.1, Revised Line 211, we add:

"We note wave-like birefringence patterns in Figure 4, and in other radargrams within this paper (Figures 5 and 6). Analogous birefringence patterns in airborne radar data have been recorded and discussed elsewhere (e.g., Gerber et al., 2023; Young et al., 2021) and have been attributed to ice sheet bulk fabric anisotropy. In principle this birefringence effect has the potential to impact the derivation of ILCI values, because it may overprint the radar reflections, effectively introducing an artefact to the data, and skewing the ILCI values. However, it is likely that the most prominent effect of this would be to overemphasise continuity in the results, which is not seen in our dataset. Here, the birefringence effect does not impact on high ILCI values associated with disrupted and buckled layers because the typical form of the birefringence pattern is substantively distinct from disrupted and buckled layers. A clear example of this is shown for the Lambert onset zone (Figure 4), where layer disruption and high ILCI values are readily apparent and do not correspond to variations in the visibility of the birefringence."

**2. Linking ILCI values with ice flow velocity**

In the paper, the authors often connect low ILCI (i.e., disrupted layering) and high flow velocity. This is one of the statements repeated very often in the paper. However, my impression from the results of the analyses is that other specific correlations are the much more useful background information related to the disturbed stratigraphy and these should be brought to the foreground. It is true that, especially in Zone 2, the low ILCI is associated with higher flow velocity (and acceleration). However, I don't think that just an increase in flow velocity can disrupt internal layering, because it

would only lead to along-flow dilatation, which should have no other effect on the layering than thinning. My feeling is that horizontal shortening in the ice that starts in this region of convergent flow and acceleration, that leads to englacial folding, which in turn leads to backscatter loss at the steep internal layers. This leads to a low ILCI, but at this point the ILCI itself is no longer relevant but rather the fold geometry, which allows further analyses and interpretation. In my opinion, the statement that a higher flow velocity (magnitude) is related to disrupted layering is too simplistic. It coincides in this Zone but is not necessarily related to each other.

We agree completely with the reviewer's caution about ascribing changes in velocity to changes in ILCI. In our existing Section 3.2 and especially lines 126-130 we feel we have been careful to explain the different causes by which layers can become buckled. From this an association between low ILCI and enhanced or fast flow can be made (as noted, for example, by Karlsson et al., 2012).

We have maintained all mentions of an association between ILCI and velocity throughout the Results section of the paper (Section 4) because this is an objective reporting of the association in our study area. However, the reviewer's comment has made us realise that some sentences in Section 5 had inadvertently implied too simple a relation between ILCI and velocity. As such we propose to amend the following sentences to reflect the reviewer's comment on possible alternative processes for high ILCI (i.e. convergence):

Revised Line 315: "Discontinuous and buckled ice layers dominate the full ice column here, as a result of relatively high current ice-flow speeds (~50 – 150 ma-1) and/or ice converging into the Lambert Rift."

Revised Line 333: "[…] demonstrates a breakdown of the assumption that low ILCI values and discontinuous layering occur as a result of changes in bed topography, converging ice and/or enhanced ice velocity.[…]"

Revised Line 373: "[…] Although these ILCI returns suggest disrupted ice as a result of converging flow and/or increased ice velocity, this is not observed at the present-day ice surface (Mouginot et al., 2019)."

**3. The origin of flow bands**

The paper suggests that *so-called* flow bands are formed by variable basal sliding, and much of the discussion focuses on inferring basal shear stress in the region and comparing it with results from other studies of basal shear stress in other Antarctic ice streams. I am not yet convinced that the "flow bands", which to me are open cylindrical folds extending parallel to the ice flow, are caused by changes in basal shear stress. I have the impression that it is more of an assumption than a solid interpretation. The references to the literature in this context were not fully informative enough for me as to how the folds we see there are formed by variations in shear stress at the base of the ice. A long part of the discussion is then based on this interpretation, which I find problematic if this interpretation is not supported by more arguments.

My suggestion would be, on the one hand (if the authors keep this explanation as the most likely explanation for the formation of these folds), to explain in much more detail how differences in friction at the base produce exactly these folds (with fold axes parallel to ice flow). More references would also be helpful here, especially clearly referencing the mechanism and how it relates to the folds we see.

Secondly, an alternative explanation that the folds we see in Zone 2 in the graben System are caused by convergent flow leading to horizontal shortening (Bons et al., 2016) should be included. My impression is that the graben system itself is causing the convergent flow, as it is the only way for the ice to flow towards the coast. This mechanism would be independent of the basal properties and

would fit particularly well with the alignment of the fold axes. In my opinion, it would be helpful to show flow lines in combination with the ILCI values and the fold axes (here called flow bands). The acceleration of the ice flow would compensate for the mass gain due to the convergence and only lead to the dilatation of the folds along flow, which should have no influence on the folds shape and only reduces the amplitudes. For me, this hypothesis is even more plausible than the theory listed in the paper. In any case, arguments should be listed that speak for or against the respective theory.

Bons, P. D., Jansen, D., Mundel, F., Bauer, C. C., Binder, T., Eisen, O., Jessell, M. W., Llorens, M.-G., Steinbach, F., and Steinhage, D.: Converging flow and anisotropy cause large-scale folding in Greenland's ice sheet, Nature communications, 7, 1-6, https://doi.org/10.1038/ncomms11427, 2016

We thank the reviewer for sharing our interest in the flow-band features and for providing a viable alternate interpretation. The insightful comments and theories highlight how little we, as a glaciological community, know about the formation of these englacial features. We suggested basal shear stress as an option for the formation of the englacial folds because the extent of the radar survey limits our ability to trace further upstream the folds that we found in the onset zone. However, it is possible that a number of different processes could have produced the folds, including basal shear stress variation, flow convergence and/or potentially flow over bedrock bumps and basal melting. It is most likely that the folds were formed through a combination of flow convergence and changes in basal boundary conditions (e.g. from a frozen bed to a sliding bed) as the ice converges into the Lambert Rift. The variations in basal boundary conditions are responsible for the folds having different wavelengths, but it is likely that the folds are then exaggerated by convergence.

We have added the idea suggested by Reviewer 1 that convergent ice flow is a viable alternative for the formation of the flow band buckling we see in Lambert Glacier throughout the revised manuscript. Although not as clearly defined in the data, the buckles are apparent up-ice of the onset zone (>15 ma$^{-1}$) and tend to propagate along flow. We agree with the reviewer that, alongside changes to the basal stresses, it is very likely that converging ice has an impact on the formation of englacial folds. To obtain a better understanding of formation of these features, it would be necessary to include bed roughness, strain rate analysis and extensive modelling scenarios, but we suggest that that is beyond the scope of the current manuscript. Further details on how we have responded to the reviewer's comments are provided via the specific points below.

We propose to add:

Revised Line 320: "We propose that variable basal sliding, in combination with converging ice flow and increased ice-flow speeds across Zones 2 and 3, are the primary cause of low ILCI values in the region (which qualitatively define visible internal layer buckles) (Fig. 5)."

Note that in the revised sentence above we have also changed "…high ILCI .." to "…low ILCI..." to correct an error flagged by the reviewer in the minor comments below.

Revised Line 353: "As we assume that these buckled layers are the product of differential ice motion caused by lateral shear stresses at the transition from slow to faster flow and ice the convergence into the Lambert Rift (Siegert et al., 2003),

We thank the reviewer in comments below for referring us to Wolowick et al. (2014) which we have cited in this sentence.

Revised Line 367: "[...] bands are likely to have formed as a result of differential basal conditions causing high basal sliding at the location where ice is converging (and the resultant speed up of ice flow) (Ely and Clark, 2016; Wolovick et al., 2014)."

**Specific points**

**L33-35:** I am not sure if the statement that the ILCI values are *"revealing the transition from internal-deformation-controlled to basal-sliding-dominated ice flow"* should be made (see further comments).

Thank you for this comment. We have removed this statement from the updated manuscript and replaced it with:

Revised Line 34: "….along the shear margin, suggesting a transition in englacial deformation conditions and converging ice flow. These zones are characterised by buckled and folded englacial layers which have fold axes aligned with the current ice-flow regime."

**L35:** *"which align"* → what does align, the fold axes of the folds?

Thank you for this point. The fold axis aligns. This has been changed in the revised manuscript to say:

Revised Line 35: "These zones are characterised by buckled and folded englacial layers which have fold axes aligned with the current ice-flow regime."

**L93:** What is the *"(65)"* referring to?

65 is the maximum horizontal gradient of ice velocity. The units for this were missed from the caption, therefore the updated caption includes 65 ma$^{-1}$.

**L98:** *"The RES data used for this study were acquired in 2007-2009 over the northern region..."* → over the northern region of the RES grid?

Northern region of the entire AGAP survey. We suggest that the manuscript does not need changing to provide any additional clarification.

**L98-101:** What is the vertical and horizontal resolution of the RES data?

AGAP: - Trace (horizontal) spacing (post-processed data): ~20 m - Vertical resolution: ~8.4 m

The following has been added to revised line 105-108:

"[…] system, operating over a bandwidth of 15-20 MHz, and deploying a 4 μs, 10MHz linear chirp designed to sound deeply into the ice. The radar system has a vertical resolution of ~8.4 m and a horizontal trace spacing (post-processing) for the AGAP survey of ~20 m. 2-D Synthetic Aperture Radar (SAR) processing was applied to enhance resolution and increase signal-to-noise (Frémand et al., 2022). […]"

**L113 (Section 3.2):** I think it would be good to mention that the overall quality of the radar data and the processing level have an effect on the continuity index as well as "other" potential causes, such as surface clutter (which is not the case here) or as in this case potentially the superimposed birefringence signal.

Another thing that could be added is a short statement about the benefit of this type of analysis in contrast to other more time-consuming methods, such as tracing many IRHs. For me, the advantage of

using an ILCI would be that certain information is available for a large data set, which gives first clues for the overall nature of the internal structure and where to apply further, more particular analyses.

To respond to the second point firstly, the use of ILCI to reconnoitre a large region quickly so as to identify regions to explore in more detail is what has motivated this study. We believe that we have already made the statement about the quick-look value of ILCI with revised lines 138-140, but the point has also been made well (and explored in much more detail) in Frémand et al.'s (2022) Antarctic-wide application, a reference to which we have now added to revised lines 138-140:

"ILCI therefore provides a useful mechanism for reconnoitring conditions across large regions of an ice sheet, to characterise broad ice-flow conditions and guide more detailed investigations in regions of interest (c.f., Frémand et al., 2022). It is in this spirit that we apply the ILCI technique in this study"

On the reviewer's suggestion to add more here on the effects of data quality and processing levels, we have added the following to revised line 125 – but we do not mention birefringence effects explicitly here, because the birefringence effects are instead devoted their own section later in the paper as per our proposed actions to the reviewer's main comment 1 above.

 "[…] then further averaging the trace-by-trace results over windows of a predefined length along radar tracks (Fig 2). This method employs minimal processing to avoid any dependency on filtering and background modification which may modify the englacial layer visibility."

Additionally, we have added to revised line 143:

"[…] 20% of the ice column was removed to avoid ILCI bias. This occurs where englacial layers are often absent in RES returns from deep in the ice sheet (lowermost 20%) and where surface clutter (upper 20%) may impact the results."

**L122:** add a space *"tracks(Fig. 2)"*.

This has been amended in the revised manuscript (revised line 124).

**L166:** To me, this section highlights that it is probably not an optimal idea to focus on linking the ILCI values to the magnitude of ice flow. First, because Zone 4 would be an exception (which, to be fair, is discussed later in the discussion) and second, because not the magnitude, but the pattern of flow seems to be the controlling factor in combination with the bed topography, the quality of the radar data and superimposed features, such as the birefringence signal.

What I think would be extremely helpful is to plot flow lines or some other marker for horizontal shortening (90° to ice flow), in particular at the onset of Zone 2, because it looks like the ILCI is rather a signature for convergent flow (which I would expect at the trough onset) creating buckled folds and where buckled folds are transported downstream.

We have responded to this in the reviewer's major point 2 (see above). This section of the manuscript (4.1) has been adapted to suggest other mechanisms that might cause the ILCI values. This includes stressing the impact of converging flow causing low ILCI returns in Zone 2 (see response above).

**L188:** *"zone 1"* → Zone 1

Thank you. This has been amended in the revised manuscript (revised line 198).

**L200-201:** The ILCI values for Zone 4 are really surprising and confusing to me. When I have a look at Figure 7, which is located in this zone, the stratigraphy looks very continuous but the values are extremely low. Hence, there is something in this data that falsely gives low ILCI values. At this point, I would not conclude that the radar stratigraphy in this region suggests *"disrupted flow"*. If so, what would actually be *"disruptive flow "* ?

We have discussed the explanation for the unexpected continuous stratigraphy/low ILCI in Section 5 of the paper. The issue being flagged here is that we have inadvertently implied a different interpretation in this line through poor phrasing (and that anyway it was premature, being situated in the Results section).

For the revision (revision line 210), we have dropped the end of this sentence ", suggestive of more disrupted ice flow" – and, in this way, only the unusual result is objectively reported here in the Results section, and our interpretation of it remains in Section 5 when discussing Figure 7.

**L216-217:** *"The spatial correspondence between ice flow exceeding 15 ma$^{-1}$ and the marked shift in ILCI values in Zone 2 (Fig. 3a) suggests that the onset of ice flow is important for the englacial structure of Lambert Glacier."*
→ I agree, but the question is what causes the change in ILCI (which is caused in this zone by the layer buckling, correct?). For me, it looks like this is also the region where ice flow would converge (maybe this could be checked), regardless of the change in ice flow velocity.

Thank you for this point. Please also see previous comments regarding convergent ice flow. The manuscript has been amended here to (revised line 237):

"The spatial correspondence between ice flow exceeding 15 ma$^{-1}$ , converging ice flow into the Lambert Rift and the marked shift in ILCI values in Zone 2 (Fig. 3a) suggests that the onset of ice flow is important for the englacial structure of Lambert Glacier."

**L222:** *"fold width"* → maybe better "wavelength".

Agreed, "Wavelength" is a better description. This has been amended in the revised manuscript (revised line 244).

**L223:** *"The amplitude of folding increases with ice thickness".* → What does this mean, that larger folds are found where the ice is thicker in general or that the fold amplitude varies in the ice column and increases with depth?

The fold amplitude varies in the ice column and increases with depth. The amended manuscript states (revised line 244):

"The amplitude of folding varies in the ice column and increases with depth".

**L225-229:** I think the structures you are referring to are not the fold axes but the axial traces (the line that connects the hinges of the synclines or anticlines in your radargrams vertically), or if combined to a plane, the axial planes: see https://cdn.eduncle.com/library/scoop-files/2022/7/can_image_1657899700300.jpg).

The fold axis would be oriented "horizontally" by combining the hinge points of the same anticline or syncline from one radargram to the next one, hence more or less parallel to the direction of ice flow.

We agree with this point, we are referring to the axial traces. This has been changed in the revised manuscript:

Revised Line 246-251:

"[…] propagate down-ice as the ice flows through the subglacial rift valley beneath Lambert Glacier. Although the fold axial traces are typically vertical in orientation, the lowermost parts of some diverge from the vertical close to the bed (e.g. at depths of ~2.5 km). An example of this can be seen approximately 25-30 km along RES transect A46B-A46B' and A54B-A54B' (Fig. 5) where the fold axial traces deflect towards the centre of the valley. In each case, the deflection of the fold axes appears just above a zone of basal ice that is characterised by few or no internal reflections (i.e. an 'echo-free zone')."

**L233:** Here, it says *"fold bands"*, should it be "flow bands"? If not, what are fold bands?

This was a typo, thank you for noticing. This should be flow bands, this has been changed in the revised manuscript (revised line 254).

**L236:** *"high-frequency"* → maybe better "short-wavelength"?

Agreed, short-wavelength is a better description. This has been changed in the revised manuscript (revised line 261).

**L243:** *"The fold runs parallel to ice flow"* → Do you mean "The fold axis runs parallel to ice flow"?

Thank you. "The fold axis runs parallel to ice flow" has been added into the revised manuscript (revised line 268).

**L243-244:** *"[...] in an area of accelerating and converging ice-flow velocities (~ 15 ma-1 to ~ 50 ma-1)."* → What are "converging ice-flow velocities"? What I think is meant is that ice flow is accelerating (from 15 to 50 ma$^{-1}$), and at the same time, ice flow is convergent. So, ice flow is indeed convergent there, which would support the fold formation hypothesis from Bons et al. (2016). I'm not saying that it should be done, but something that would be very interesting to see is how much horizontal shortening (90° to ice flow) along one of the fold axes (possibly the central one) is actually happening and if that fits to the change in wavelength of the folds along ice flow.

We agree. We have clarified by changing the manuscript text to say (revised line 268):

"[...] in an area of accelerating and converging ice-flow (where velocity ranges from ~ 15 ma$^{-1}$ to ~ 50 ma$^{-1}$)."

We agree that this would be interesting but the birefringence seen in the radar limits the confidence in calculating horizontal shortening across the fold axes because it would be difficult to accurately determine the fold axis and trough.

**L245-247:** *"Assuming that the fold formed and then advected down-ice with flow, we estimate that this englacial fold could have persisted for at least 10.5 ka (based on calculations of the average current ice velocity (Mouginot et al., 2019) and the distance between each fold on individual flightlines)."*

→ This is an excellent approach to investigate the stability of the system. It would be interesting to specify (in one of the maps) where you define the starting point of fold formation. Would it just be the

location where in the radargrams the fold appears for the first time, hence the southernmost of the yellow dots in Figure 6a?

Thank you. We define the starting point of the fold as the southernmost yellow dot in Figure 6a because this is the southernmost radargram, it is likely that this is not the initial formation point, however we cannot visualise the fold beyond this point. We have added the following sentence to revised line 273:

"[…] the distance between each fold on individual flightlines). It is likely that the fold extends further inland and we cannot visualise the exact location of the fold formation from existing data. We define the initiation point of the fold as the southernmost yellow dot (Figure 6a), corresponding to the southernmost radargram of the AGAP-N survey."

**L250:** Maybe also add Siegert et al. (2004) here.
Siegert, M. J. et al. Ice flow direction change in interior West Antarctica. Science 305, 1948–1951 (2004)

Thank you. Siegert et al. (2004) has been added to the revised manuscript (revised line 278).

**L267-269:** Could you specify which of the zones you are referring to?

"Zone 1". This has been added into the revised manuscript (revised line 295).

**L269-270:** *"This low ILCI in the upper ice-stream catchment is not associated with current enhanced or fast ice flow".* → If this refers to Zone 4, it says something different than in the statement in L200-201 (disruptive flow).

This statement is referring to Zone 1. The manuscript has been changed to say (revised line 296):

"This low ILCI in the upper ice-stream catchment (Zone 1) is not associated with current enhanced or fast ice flow (as the ice currently flows at <5 ma$^{-1}$)(Fig. 3),"

**L269-271:** Is it possible to refer to a Figure to see which region and radargrams are referred to?

See previous response to L269-270 referring to Zone 1 in Figure 3 (revised line 296).

**275-277:** *"Second, it is possible that disruption of the layers giving rise to low ILCI is a result of power loss from dipping englacial reflectors, as seen elsewhere in Antarctica (Holschuh et al., 2014; Winter et al., 2015)."* → Has this been checked, or is it an assumption?

The sentence being queried here is the second (hence the sentence begins "Second…") of three possible ways we can think of that might cause low ILCI values in this locale. It is not, unfortunately, something we can easily check over the spatial scales over which the data span. We think that in the context of the paragraph – posing alternate possible explanations – no action has been taken in terms of adding further text here.

**279-283:** The last of the three possibilities (like all three actually) have been listed here as potential reasons. Is it possible to look at the radar data to determine which of the three causes the low ILCI? Are the ice packets, as in Bell et al., 2011 to be found here or not? Here again I see the problem that we do not learn much from the ILCI values per se. If something is to be learned about the englacial architecture, it should be checked or excluded what is causing the low values.

We concur with the overall principle that ILCI's main value is in providing a spatial overview and identifying regions for more dedicated focus (as we have now clarified further in response to comment Line 113 above (revised lines 138-140)). That is really our main objective with this manuscript, and hence it is beyond the scope of this contribution to explore the radar data much more comprehensively in the manner implied here. It is, however, within our plans for future work on this dataset and region. Ice packets (i.e. zones of basal ice from refreezing or ice deformation) are found within the region (Bell et al., 2011), but as we are removing the lower 20% of the ice column, it is unlikely that such ice packets at the bed will be impacting the ILCI.

**L293-295:** *"We propose that variable basal sliding, and therefore variable ice-flow speeds across zones 2 and 3, are the primary reason for high ILCI values in the region (which qualitatively define visible internal layer buckles) (Fig. 5)."*
→ Shouldn't it be rater "low" values instead of "high" in these zones?

I don't fully understand how variable basal sliding in these zones would create the observed folds. If the vertical velocity profile changes along flow, you may end up with something like the moving patches of high and low resistance as proposed by Wolovick et al., 2014? This would create folds (under particular conditions), which would, however trend (the fold axis) 90° to ice flow. But the buckled folds observed here trend parallel (the fold axis) to ice flow. Also, it is proposed later that the change in basal shear strength is rather gradual than abrupt.

If this interpretation on how the folds in Zone 2 are formed is the only one presented here, there is more explanation needed as well as references on how variable basal sliding would create the observed folds. The explanation that I would add as an equally well possibility is that the folds are created via horizontal shortening due to the flow convergence (Bons et al., 2016) when the ice is flowing into the main trough. This would match the orientation of the fold axis and would be independent of processes at the ice base.

Wolovick, M. J., T. T. Creyts, W. R. Buck, and R. E. Bell (2014), Traveling slippery patches produce thickness- scale folds in ice sheets, Geophys. Res. Lett., 41, 8895–8901, doi:10.1002/2014GL062248.

Thank you for pointing this out. This should be low values rather than high ones.

This comment has highlighted that the previous version of the manuscript has incompletely considered the significant impact that converging ice flow will have on the formation of englacial folds and consequential low ILCI, and this has been added into the revised manuscript.

We believe that a number of different processes could have produced the folds, including basal shear stress variation, flow convergence and/or potentially flow over bedrock bumps and basal melting. It is most likely that the folds were formed through a combination of both flow convergence and changes in basal boundary conditions because of the transition from grounded ice as the ice converges into the Lambert Rift. We believe that the basal boundary conditions cause the buckles to have different wavelengths and cause the deflection of the fold axis in the lower ice column (Fig. 5).

We have included this as a potential cause in the revised manuscript but would also like to include variable basal conditions as a potential formation. We have revised the text to state (revised line 321):

"We propose that variable basal sliding in combination with convergent ice flow and increased ice-flow speeds across Zones 2 and 3, are the primary cause for low ILCI values in the region (which qualitatively define visible internal layer buckles) (Fig. 5)."

**L305-312:** Maybe I need clarification, but in the first paragraph of Section 5.1 you proposed three possibilities for the low ILCI values (assuming that in section 5.1 you were referring to this zone). But here, only the attenuation is mentioned. How does this fit together?

The revised manuscript clarifies that power has not been lost through dipping reflectors or basal ice units. A plausible possibility for low ILCI here is therefore attenuation. Revised line 338 will be amended to:

"[…] the radar energy, generating weaker reflections with implications for the ILCI analysis. Because 20% of the lower ice column is removed from the analysis, it is unlikely that basal ice units will impact the ILCI values. We therefore suggest that thicker ice causing weaker reflections is the most plausible cause of low ICLI values.[…]"

**L319-320:** *"As we assume that these buckled layers are the product of lateral shear stresses at the transition from slow and fast flow (Siegert et al., 2003a), [...]"*

→ I had a look at Siegert et al., 2003 and tried to understand the argumentation chain: onset of fast flow -> variable lateral shear stresses --> leads to layer buckling as observed here. However, Siegert et al., 2003 write:
"It should be noted that no attempt is made to quantify the degree of layer buckling or determine the mechanics responsible for "buckled" internal layering. It is likely that buckled layers occur as a consequence of high longitudinal stresses within regions of enhanced ice

flow, and lateral shear stresses at the transition of fast- and slow-flowing parts of the ice sheet (Jacobel et al., 1993). The assumption made in this paper is that these stresses occur as a consequence of enhanced ice flow, and that internal layers will become more buckled the longer such stresses are applied."
Jacobel et al., 1993 discuss a lot of possibilities for their buckled folds, but I have the feeling that at the end there is no clear conclusion that basal shear stresses are responsible for the folds they observe. They also state: " Certainly, the shorter wavelength of the folds on either side of the bedrock rise is suggestive of greater lateral compression there."
→ which points towards the horizontal shortening theory.
I did not investigate the origin of the basal shear stress theory further and I had the impression that variations in basal shear stress is rather an assumption and one of many possible explanations. If there is literature that allows a clear connection to be drawn between the folds observed here and the basal shear stress approach, it would be good to discuss it here.

From our own reading of the literature, and the reviewer's comments, we have reached a conclusion that there appears to be a fundamental lack of understanding, or at least something of an opaqueness, of the process of how such buckled layers are formed. Jacobel et al. (1993) used simple modelling to show that folding of internal layers could be directly related to bedrock or variations in basal conditions. Karlsson et al. (2009) suggested that variations in sticky basal conditions could have caused similar features at Pine Island and although Siegert and other (2003) made no attempt to determine the formation of buckling, basal conditions were considered in conjunction with enhanced ice flow. Similar suggestions for fold formation were suggested by Ng and Conway (2004) who again stressed the importance that "*Although the origin of the folds in unknown, a simple analysis shows that they carry an important paleostreaming signature.*" Because our manuscript does not set out to investigate the origin of the folds in Lambert Glacier, we have simply suggested possible formation mechanisms. To obtain a better understanding of these features, it would be necessary to include bed roughness, strain rate analysis and extensive modelling scenarios, which we feel is beyond the scope of the current manuscript.

In line with reviewer 1, we do suggest that a greater case for compressional forces associated with convergent ice flow should be put forward as a suggestion for buckle formation throughout the text of the manuscript. Specifically, we have changed revised line 354 to:

"As we assume that these buckled layers are the product of lateral shear stresses at the transition from slow and fast flow combined with the impact of converging ice flow (Siegert et al., 2003), [...]"

*Karlsson, N. B., Rippin, D. M., Vaughan, D. G., and Corr, H. F.: The internal layering of Pine Island Glacier, West Antarctica, from airborne radar-sounding data, Annals of Glaciology, 50, 141-146, https://doi.org/10.3189/S0260305500250660, 2009.*

*Ng, F. and Conway, H.: Fast-flow signature in the stagnated Kamb ice stream, West Antarctica, Geology, 32, 481-484, 2004.*

**L333-334:** *"[...] therefore the flow bands are likely to have formed as a result of differential basal conditions causing high basal sliding (and resultant ice flow speed up) [...]"*
→ My comment here is a repetition of previous comments that it is unclear to me how the change in basal properties leads to the folds in Zone 2.

This comment links back to the previous comment: We have emphasised convergent ice flow as a potential cause more widely throughout the revised manuscript. We have revised the statement to say (revised line 367):

"[...] therefore the flow bands are likely to have formed as a result of differential basal conditions causing high basal sliding at the location where ice is converging (and the resultant speed up of ice flow) (Ely and Clark, 2016; Wolovick et al., 2014)."

**L421-422:** *"The englacial stratigraphy in Zone 2 demonstrates a gradual (rather than abrupt) transition from internal deformation to basal sliding at the onset zone."*
Again, a repetition of previous comments that it is not clear to me how the change in basal properties leads to the folds in Zone 2 and I am also not convinced that the folds in Zone 2 demonstrate this transition. If so, the relationships should be explained precisely and more clearly.

Reviewer 2 made a similar comment. We begin to see englacial disruption before the onset of faster ice (e.g. around the black line noting 15 ma$^{-1}$). This velocity contour description has been added to the figure caption on Fig 3a.

**L447-449:** It has already been partly addressed in previous paragraphs, but for me, a conclusion of this study is that the ILCI values alone are not very meaningful. The interesting thing is what they say about radar stratigraphy when looking at low and high continuity zones. Low values can indicate folded layers, but also low resolution of the radar system, poor visibility of the layers due to high attenuation or being influenced by signals in the radar data, which are not just internal layers.

We agree with this comment. We have added the following to revised line 485:

"[…] The findings also highlight the limitations of using the ILCI method in isolation and we recommend that these limitations be taken into account when the ILCI method is applied in future. […]"
* * *
Reviewer 2

**General Comments** Sanderson and others present interesting and original work focused on deriving a very detailed picture of the englacial layering within the Lambert Glacier in East Antarctica. We recognize the amount of time and effort required to trace such a huge volume of radar data, and this

will be a key asset for the AntArtchitecture SCAR project as well as for large-scale modeling efforts that are learning to invert for englacial layering. There are several novelties in this work, including the mapping of a large-scale englacial buckling feature over 100s of kilometers across many radar transects, and its mapping with surface topography and ice flow velocity, as well as the interesting point that a low ICLI does not necessarily imply unmappable englacial reflectors, which opens up the potential for a subtler use of the ICLI in many other glacier catchments…? The figures are all very impressive with a lot of information and the use of 3D effects for the radargrams and basemap views are very helpful to better understand the flow dynamics. We applaud the authors for this. We recommend this manuscript for publication after some minor revisions outlined here below.

We would like to thank Dr Marie Cavitte for her insightful comments which have helped us to improve the manuscript. Our responses to each of the minor comments raised, and how we have addressed them for the revised version of the manuscript, are outlined below. For this, reviewer comments are copied verbatim in blue, and our response to each is given in black.

**Specific comments**

-In the discussion section, I found myself flipping a lot between fig 1 to have the ice thickness, bed topography and general ice flow speed geometries to compare to fig 3 where each zone with discussed in details. Would it be possible to perhaps add some ice thickness contours or ice velocity contours, or have one of the two panels overlying ice thickness/ice flow velocities so we don't have to flip between the two figures? It would make it easier to really see for ourselves what is discussed.

In particular, it made it hard for me to see the gradual transition discussed in Section 5.2. For me, the transition shown in Fig 3 is quite abrupt but perhaps I am missing where the "onset region" is referring to? Could this be labeled on figure 3, as was the tributary T ?  Same point L338-339 where I need the ice velocities to be mapped on Fig 3 so I can see where the ILCI returns map with the ice velocities.

Thank you, we appreciate the difficulties of comparing figures highlighted by the reviewer. The velocity contour for ice flow >15 ma$^{-1}$ is actually on figure 3a but it is not labelled or mentioned in the caption. This was an oversight on our part and the caption has been amended to include this.

-In Section 4.2, has the fact that the ICLI gives low values yet englacial layering is still laterally continuous (i.e. be traced) been highlighted in previous studies? If not, I think it's something that should be highlighted more strongly here, for impacts in other (past, future) studies.

Thank you for suggesting that we should highlight this point. We think it makes most sense to make this point in the Discussion so have added the following to revised line 342:

"[…] continuous englacial layers. A broader point relevant to wider Antarctic studies beyond the Lambert region studied here, is that englacial layers in Zone 4 remain clearly traceable in the radar data despite generating low ILCI (Figure 7). This demonstrates that while ILCI typically acts as a useful first filter for assessing layer continuity (c.f. Frémand et al., 2022), prospects for tracing layers should not be dismissed across low ILCI zones."

-The order in section 5.1 of the Discussion is a little confusing as the three likely causes for the low ILCI values are discussed at the very start and the evidence is only given in the 3$^{rd}$ paragraph starting L305. Could the order be revised a little?

The omission here is that we had not made clear that the first paragraph was interpreting causes for low ILCI *in Zone 1*. The 2$^{nd}$ paragraph then interprets patterns in Zone 2 and 3, and the 3$^{rd}$ paragraph queried by the reviewer above is our interpretation *specific to Zone 4*.

We have made it clearer in the revised Section 5.1 by more explicitly naming Zones 1…4 in the writing.

-Will the englacial reflectors be published with this manuscript ? There is no mention of this in the Data availability section.

The ILCI values and the englacial reflectors along the shear margin will be published alongside the manuscript. We have added a statement in the Data Availability section and included a DOI to link the data.

**Technical comments**

Abstract

L31 – I would not use inverted commas around *internal-layering continuity index*.

Agreed and this has been amended (revised line 31).

Introduction

L44 - "...lower **in** magnitude…"

Thank you for noticing, this has been amended (revised line 44).

L46 – I would suggest to cite the latest IPCC AR6 report here too.

Agreed, this citation has been added to the revised manuscript (revised line 48).

L69 – For the accumulation rates reconstructions, suggestion to cite also *Cavitte et al, 2018, The Cryosphere?*

Thank you. This citation has been added to the revised manuscript (revised line 69).

Methods

L104-105 – I would change "reduce echo signal noise" to "increase signal-to-noise".

Agreed and this has been amended (revised line 107).

L135 – it is stated that englacial layers "are often absent in RES returns", however, I think this is specific to how the radar system is set up. You can set up to get shallow layering with a high gain channel. And snow radar can also capture surface layering. I would suggest to be more specific, by adding "in **deep** RES returns" ?

Thank you for the suggestion. The revised manuscript now states (revised line 150):

"This occurs where englacial layers are often absent in RES returns from deep in the ice sheet (lowermost 20%)….."

Discussion

L195 – add "likely" in front of "the cause of low layer…" and maybe replace "layer" here by "reflector" to be consistent with your naming convention? Although I understand here it is referring to the ILCI…

Thank you for the suggestion. The revised line 205 of the manuscript and included:

"the likely cause of the low reflector…"

L226 – It might be more interesting to say at what height above the bed you are here, instead of how deep (2.5 km depth)?

Thank you for the suggestion. This has been changed in the revised manuscript (revised line 248):

"some diverge from the vertical close to the bed (e.g. within 1 km above the bed)."

L228 – The deflection of the fold axes is not visible or marked on fig 5, is it? It would be nice to highlight it.

Agreed. The figure has been edited and added makers where the fold axial deflects.

L399 – Adding the Gerber et al., 2021 paper here would be great! https://tc.copernicus.org/articles/15/3655/2021/.

Agreed, this citation has been added to the revised manuscript (revised line 434).

Figure 1 – It's hard to see the yellow circles on panel a. Also, i printed form, in panel c, we don't see the underlying raster of maximum horizontal gradient. And what does the "65" in brackets refer to each time? Can you explain somewhere how this raster is derived? It is used in several figures yet it is unclear to me.

Yellow circles have been enlarged. Panel c has been re-exported at a higher quality to capture the underlying raster. 65 is the maximum horizontal gradient of ice velocity. The caption has been edited to include:

"…Ice-flow speed averaged over 1992-2017, overlying a raster derived from calculating the slope from the maximum horizontal gradient of ice velocity. This underlying raster has a grey-scale colour pallet and is histogram equalised with the colour scale saturated at 65 ma-1 (Mouginot et al., 2019)."

Figure 2 – the caption mentions panel (d) but the figure itself has panel (e) marked ? Are the red unsmoothed ILCI results really useful on panels a-c? I feel like they could be removed to declutter a little the radargrams (and they are difficult to see at this scale), up to the authors.

Thank you for spotting this, panel (d) was missed. This has been amended so that the current (e) with become (d) in the revised manuscript. We understand that the radargrams may appear cluttered, however the red unsmoothed ILCI results are useful to demonstrate the trace by trace aspect of the calculation.

Figure 4 – the radargram is very washed out in printed format. Is it possible to increase the contrast? Also, the red curve and the brown shading of the bedrock should be mentioned in the figure caption. If it corresponds to the velocity, it might be clearer if the velocity magnitude axis were also written in red.

Thank you for this point. We have edited the figure to increase the contrast, change the velocity magnitude axis to red and include the descriptions of both the red curve and bed rock shading in the caption.

Figure 5 – there is some striping on the radargrams that makes it difficult to see the layering well. Could the radargrams be enhanced somehow? I understand this is always tricky. Panel d is missing the little d on the panel itself, and the underlying raster is not described in the caption.

We believe that the "striping" is a result of the birefringence effect on the radargrams (see comments from Reviewer 1). Rather than enhancing the radargrams, we have added the following sentence to the caption:

"Birefringence patterns visible in the radargrams (a,b, and c) are discussed in Section 4.1"

(d) has been added to panel d and a description of the underlying raster has been added to the caption.

Figure 7 - Why is Figure 7 so far down in the paper, and not close to figure 4?

We included Figure 7 within the text of discussion to demonstrate the low ILCI values against continuous layering. The figure directly relates to the text above it and isn't referred to before this point.